# Fermi polaron laser in two-dimensional semiconductors

Tomasz Wasak,[1] Falko Pientka,[2, 1] and Francesco Piazza[1]

[1]*Max-Planck-Institut für Physik komplexer Systeme, Nöthnitzer Str. 38, 01187 Dresden, Germany*
[2]*Institut für Theoretische Physik, Goethe-Universität, 60438 Frankfurt am Main, Germany*

We study the relaxation dynamics of driven, two-dimensional semiconductors, where itinerant electrons dress optically pumped excitons to form two Fermi-polaron branches. Repulsive polarons excited around zero momentum quickly decay to the attractive branch at high momentum. Collisions with electrons subsequently lead to a slower relaxation of attractive polarons, which accumulate at the edge of the light-cone around zero momentum where the radiative loss dominates. The bosonic nature of exciton polarons enables stimulated scattering, which results in a lasing transition at higher pump power. The latter is characterized by a superlinear increase of light emission as well as extended spatiotemporal coherence. As the coherent peak is at the edge of the light-cone and not at the center, the many-body dressing of excitons can reduce the linewidth below the limit set by the exciton nonradiative lifetime.

## I. INTRODUCTION

Atomically thin semiconductors like monolayer transition metal dichalcogenides (TMDs) [1, 2] exhibit a series of interesting optical properties [3–12] and provide promising platforms for the development of useful photonic devices [13–17]. An essential feature for optical control is the existence of tightly bound excitons. Moreover, monolayer TMDs allow for electrical injection of itinerant electrons, which transforms excitons into exciton-polarons [18–20], i.e., the optical response is governed by excitons dressed by the electronic bath forming attractive and repulsive Fermi-polaron branches [21–32]. Residual interactions between exciton-polarons mediated by the itinerant electrons can cause strong optical nonlinearities with an effective strength largely exceeding the direct interaction between the tightly bound excitons [33–35]. Besides its relevance for nonlinear optics, polaron formation can induce intriguing collective phenomena both in [36] and out of equilibrium [37, 38].

Motivated by this progress, we theoretically study the nonlinear relaxation dynamics of optically excited exciton-polarons. Specifically, we consider the situation depicted in Fig. 1(a), where excitons are pumped into the higher-energy repulsive polaron branch. Subsequent relaxation into finite momentum states of the lower branch creates a metastable population of attractive polarons, which decays either through slow nonradiative processes or through fast radiative processes limited to very low momenta within the light cone. A key result of this paper is that the relaxation dynamics of attractive polarons change qualitatively as a function of pump power. At low power, relaxation to near-zero momentum states happens through a cascade of scattering events with small momentum transfer, creating a bottleneck that makes the conversion of attractive polarons into photons inefficient [black arrows in Fig. 1(a)]. At high pump power, however, polarons accumulate just outside the light cone, where relaxation is particularly slow. The large occupation of low-momentum polaron states triggers stimulated scattering to this region in momentum space [red dotted arrow in Fig. 1(a)], which short-circuits the cascade and dramatically enhances the radiative efficiency as well as the spatial and temporal coherence of the emitted light.

In contrast to a cavity-enhanced coupling, where the population is accumulated at the center of the light-cone [39–42], here the light-matter coupling is too weak to form exciton-polaritons that would enable relaxation to zero momentum. While this obviously reduces the emission intensity, it also allows for a potentially smaller linewidth as the dressing of excitons by electron-hole excitations can increase the polaron lifetime beyond the nonradiative exciton lifetime. This situation is analogous to bad-cavity lasers, where the excitation is mainly stored in the gain medium [43].

The bottleneck enabling the lasing transition originates from the small scattering phase space available for low-momentum polarons. This is a generic feature of equilibration, suggesting a transition can exist independent of the relaxation mechanism such as scattering by electrons, phonons, or disorder. Below, we specifically consider relaxation by electron-exciton collisions.

## II. POLARON KINETIC EQUATION

Excitons interacting with itinerant electrons in 2D can be described by the Hamiltonian

$$\hat{H} = \hat{H}_x + \hat{H}_e + \hat{H}_{\text{int}}, \qquad (1)$$

where $\hat{H}_x = \sum_{\mathbf{k}}(\mathbf{k}^2/2m_x)\hat{x}_{\mathbf{k}}^\dagger \hat{x}_{\mathbf{k}}$ and $\hat{H}_e = \sum_{\mathbf{k}} \varepsilon_e(\mathbf{k})\hat{e}_{\mathbf{k}}^\dagger \hat{e}_{\mathbf{k}}$ describe free excitons and electrons with $\varepsilon_e(\mathbf{k}) = \mathbf{k}^2/2m_e - E_F$. Bosonic (fermionic) annihilation operators of excitons (electrons) are denoted by $\hat{x}_{\mathbf{k}}$ ($\hat{e}_{\mathbf{k}}$). We model the interaction as an attractive contact potential, $\hat{H}_{\text{int}} = U \int d^2r\, \hat{x}^\dagger(\mathbf{r})\hat{x}(\mathbf{r})\hat{e}^\dagger(\mathbf{r})\hat{e}(\mathbf{r})$, with an effective strength given by the trion binding energy $E_B$. We choose $m_x = 2m_e$, $E_F = E_B \equiv q_B^2/m_e = 25\,\text{meV}$ throughout this paper. While a realistic interaction is more complicated, our main results mostly depend on energy scales far below $E_B$, where the contact interaction is an excellent approximation [44, 45]. The excitation spectrum of this system, which is dominated by attractive

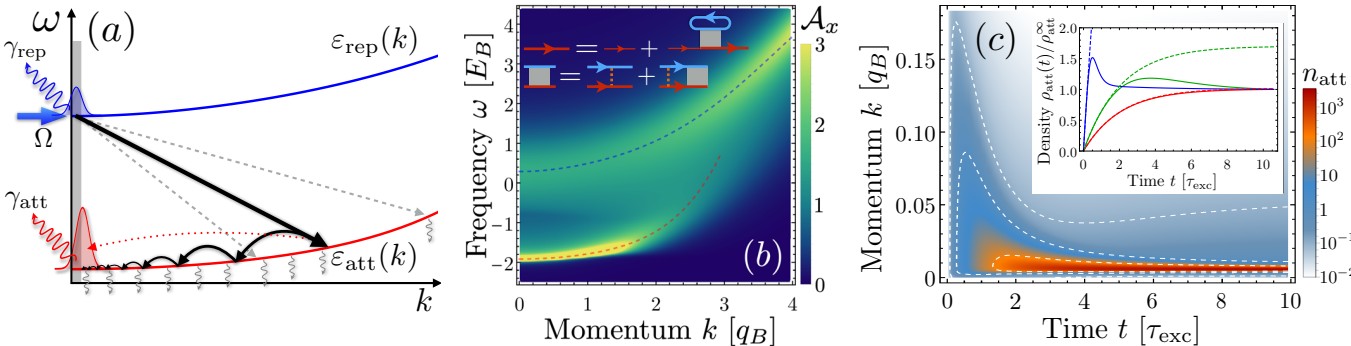

FIG. 1. (a) Polaron dynamics. Driving the repulsive branch and radiative decay of both branches only occur within the light cone (shaded region at $k \approx 0$). Repulsive polarons (blue peak) can decay into a range of momenta of the attractive branch (half-width indicated by dashed gray arrows). Attractive polarons can further be nonradiatively lost or relax to smaller momenta via collisions with electrons. A population build-up at the edge of the light cone (red peak) enables stimulated scattering to these momenta (curved dotted red arrow). (b) Exciton spectral function showing a broad repulsive and a narrower attractive polaron resonance with corresponding dispersions $\varepsilon_\alpha(\mathbf{k})$ (dashed lines). Inset: Dyson equation for the exciton Green's function (red arrow), involving electrons (blue arrow), the $T$-matrix (gray square), and the interaction $U$ (dotted orange line). (c) (Main panel) Time evolution of the distribution function at $\Omega = 10^{-6}E_B q_B^2$. The dashed white contours correspond to $n_{\text{att}} = 1/10, 1, 100$. (Inset) Attractive polaron density $\rho_{\text{att}}(t)$ at $\Omega = 10^{-8}, 10^{-6}, 10^{-4}\ E_B q_B^2$ (solid lines from bottom to top). The dashed lines indicate the approximate solution neglecting radiative decay.

and repulsive polarons separated by a broad trion-hole continuum [27], is calculated within a self-consistent $T$-matrix approximation, see Appendix A, and shown in Fig. 1(b). The driven dissipative dynamics of the polaron population can be described by a kinetic equation for the bosonic distribution function $n_\alpha(\mathbf{k}, t)$,

$$\partial_t n_\alpha(\mathbf{k}, t) = -\gamma_\alpha(\mathbf{k}) n_\alpha(\mathbf{k}, t) + \Omega_\alpha(\mathbf{k}) + I_\alpha(\mathbf{k}, t), \quad (2)$$

where the index $\alpha \in \{\text{att}, \text{rep}\}$ labels the polaron branches. This equation is formally derived within nonequilibrium quantum field theory in Appendix A employing the formalism of Green's functions. For the derivation of the relevant Green's functions and the self-energies, we refer the reader to Appendices B and C, respectively. The three terms on the right-hand side of this equation describe polaron decay, pumping, and collisions with electrons, which are all described in detail below. We assume a low polaron density allowing us to ignore collisons between polarons. A pictorial illustration of the relevant processes is shown in Fig. 1(a).

The relaxation processes mediated by the itinerant electrons are governed by the collisional integral

$$I_\alpha = \frac{1}{V} \sum_{\beta, \mathbf{q}} \left[ W_{\mathbf{k}\mathbf{q}}^{\alpha\beta} [n_\alpha(\mathbf{k}) + 1] n_\beta(\mathbf{q}) - W_{\mathbf{q}\mathbf{k}}^{\beta\alpha} [n_\beta(\mathbf{q}) + 1] n_\alpha(\mathbf{k}) \right],$$

$$(3)$$

where $\beta \in \{\text{att}, \text{rep}\}$ and $V$ is the sample area. The polaron transition matrix elements $W_{\mathbf{k}\mathbf{q}}^{\alpha\beta}$ take the form

$$W_{\mathbf{k}, \mathbf{k}'}^{\alpha\beta} = \frac{2\pi}{V} \sum_{\mathbf{Q}} \left| T[\mathbf{Q}, \varepsilon_\beta(\mathbf{k}') + \varepsilon_e(\mathbf{Q} - \mathbf{k}')] \right|^2 Z_\alpha(\mathbf{k}) Z_\beta(\mathbf{k}')$$

$$\times \delta \left[ \varepsilon_\alpha(\mathbf{k}) + \varepsilon_e(\mathbf{Q} - \mathbf{k}) - \varepsilon_e(\mathbf{Q} - \mathbf{k}') - \varepsilon_\beta(\mathbf{k}') \right]$$

$$\times n_e(\mathbf{Q} - \mathbf{k}')[1 - n_e(\mathbf{Q} - \mathbf{k})], \quad (4)$$

where $n_e(\mathbf{q}) = \theta(k_F - |\mathbf{q}|)$ is the $T = 0$ Fermi distribution and $\varepsilon_\alpha(\mathbf{k})$ denotes the polaron dispersion. This result is equivalent to Fermi's golden rule, where the transition amplitude $T(\mathbf{Q}, \omega)$ is computed in a self-consistent $T$-matrix approximation, i.e., $T$ can be understood as the Green's function of an exciton-electron pair (trion). In addition, the transition amplitudes are renormalized by the polaron quasiparticle weights $Z_{\alpha, \beta}(\mathbf{k})$, which quantify the spectral weights of the resonances in Fig. 1(b).

The exciton decay rate is given by $\gamma(\mathbf{k}) = \gamma_{\text{exc}} + \gamma_{\text{rad}}(\mathbf{k})$, where $\gamma_{\text{exc}} \equiv 1/\tau_{\text{exc}}$ is a momentum independent nonradiative decay rate (e.g., trapping by local charges) and $\gamma_{\text{rad}}(\mathbf{k})$ is the momentum dependent rate for radiative decay. The results presented below remain qualitatively valid if the hybridization between excitons and photons is negligible and if the radiative decay is maximal at $\mathbf{k} = 0$ and has a smooth behavior at the light-cone boundary. Concretely, we assume a bad cavity with a photon loss $\gamma_{\text{ph}}$ greatly exceeding the coupling strength $g$ and the detuning between the cavity and the attractive polaron resonance at $k = 0$. In this limit, the cavity remains essentially unoccupied and the annihilation operators of photons, $\hat{a}_\mathbf{k}$, and excitons are approximately related by $\hat{a}_\mathbf{k} = g\hat{x}_\mathbf{k}/(\varepsilon_{\text{ph}}(\mathbf{k}) - \varepsilon_{\text{att}}(\mathbf{k}) - i\gamma_{\text{ph}}/2)$ with $\varepsilon_{\text{ph}}(\mathbf{k}) = \delta + k^2/2m_{\text{ph}}$ the photon dispersion. Substituting this expression into the coupling term $g\hat{a}_\mathbf{k}^\dagger \hat{x}_\mathbf{k} + \text{h.c.}$, we find the real part of the exciton self-energy responsible for the formation of polaritons to be suppressed relative to the imaginary part $\gamma_{\text{rad}}(\mathbf{k}) = (g^2/2)\gamma_{\text{ph}}/[(\varepsilon_{\text{ph}}(\mathbf{k}) - \varepsilon_{\text{att}}(\mathbf{k}))^2 + \gamma_{\text{ph}}^2/4]$, and the former can be thus neglected. The mixing between excitons and photons is appreciable only within a narrow light-cone around $\mathbf{k} = 0$, setting the width of the radiative loss profile. The control of the radiative decay of excitons in TMDs via the encapsulating material

has been recently demonstrated [46]. As the polaron is a composite particle, its decay rate $\gamma_\alpha(\mathbf{k}) = Z_\alpha(\mathbf{k})\gamma(\mathbf{k})$ is suppressed by the quasiparticle weight, which measures its excitonic content. A similar suppression applies to the drive $\Omega_{\rm rep}(\mathbf{k}) = Z_{\rm rep}\Omega(2\pi)^2\delta^2(\mathbf{k})$ modeled as continuous pumping of the repulsive polaron at $\mathbf{k} = 0$ with strength $\Omega$, where $Z_{\rm rep} = Z_{\rm rep}(k=0)$. We choose typical parameter values for our numerics (see [33, 47–49]): $g = 0.00709E_B$, $m_{\rm ph} = 10^{-4}m_e$, $\tau_{\rm exc} = 0.22$ ns ($\gamma_{\rm exc} = 1.2\times10^{-4}E_B$), $\gamma_{\rm ph}^{-1} = 0.557$ ps ($\gamma_{\rm ph} = 0.0473E_B$), and zero detuning $\delta = \varepsilon_{\rm att}(0)$.

## III. RELAXATION DYNAMICS

In TMDs at cryogenic temperatures we have $E_B \simeq 10^2 T$ and we henceforth set $T = 0$, thereby ignoring transitions from the attractive to the repulsive branch. The kinetic equation for the latter reads

$$\dot{\rho}_{\rm rep}(t) = -\Gamma_{\rm rep}\rho_{\rm rep}(t) + Z_{\rm rep}\Omega - Z_{\rm rep}\gamma(0)\rho_{\rm rep}(t), \quad (5)$$

where $\rho_\alpha = (1/V)\sum_{\mathbf{k}} n_\alpha(\mathbf{k})$ is the polaron density. The effective decay rate $\Gamma_{\rm rep}[n_{\rm att}] = (1/V)\sum_{\mathbf{k}}[1 + n_{\rm att}(\mathbf{k},t)]W_{\mathbf{k},\mathbf{k}'=0}^{\rm att,rep}$ is set by the transition rate. Due to rapid relaxation of attractive polarons, their occupation in the relevant high-momentum region remains small for all pumping strengths and we can set $\Gamma_{\rm rep}[n_{\rm att}] \approx \Gamma_{\rm rep}[0] \equiv \Gamma_{\rm sc}$. The characteristic scale for this relaxation rate is $E_B$, which we assume to greatly exceed the radiative decay rate $Z_{\rm rep}\gamma_{\rm rad}$, such that essentially all repulsive polarons decay to the attractive branch. Ignoring the radiative decay yields a solution of Eq. (5) equal to $\rho_{\rm rep}(t) = \rho_{\rm rep}^s(1 - e^{-\Gamma_{\rm sc}t})$. The stationary density $\rho_{\rm rep}^s = Z_{\rm rep}\Omega/\Gamma_{\rm sc}$ grows linearly with pump strength and is reached on very short timescales $\sim \Gamma_{\rm sc}^{-1}$.

Attractive polarons are initially generated at high momenta of order $q_B$ which is of order $k_F$ here, and subsequently relax to lower momenta. This dynamics is described by Eq. (2), which for our choice of pump and at $T = 0$ simplifies to

$$\dot{n}_{\rm att} = \tilde{I}_{\rm att} + P_{\rm att} - \gamma_{\rm att}n_{\rm att}, \quad (6)$$

where we suppressed momentum and time variables and $\tilde{I}_{\rm att}$ only contains scattering within the attractive branch. The transitions from the repulsive branch appears via the effective pump $P_{\rm att}(\mathbf{k},t) = W_{\mathbf{k},\mathbf{k}'=0}^{\rm att,rep}[1 + n_{\rm att}(\mathbf{k})]\rho_{\rm rep}(t) \approx W_{\mathbf{k},\mathbf{k}'=0}^{\rm att,rep}\rho_{\rm rep}(t)$. In the remainder of the paper, we calculate the attractive polaron dynamics by solving Eq. (6).

We first analyze the attractive polaron density $\rho_{\rm att}(t)$ shown as solid lines in the inset of Fig. 1(c) for various pump strengths $\Omega$. An initial quadratic growth on a timescale $\sim E_B^{-1}$ (not visible) turns linear until the timescale of the exciton nonradiative lifetime is reached. The subsequent dynamics is qualitatively different depending on the pump strength. For weak pumps, the density monotonically approaches its final value. In this

case, most attractive polarons eventually decay nonradiatively and collisions within the attractive branch do not influence their density, which can be approximated by $\rho_{\rm att}(t) = (Z_{\rm rep}/Z_{\rm att})\Omega\tau_{\rm exc}[1 - \exp(-Z_{\rm att}\gamma_{\rm exc}t)]$ when we ignore dynamics on short timescales $\sim \Gamma_{sc}^{-1}$ and replace the spectral weight by a constant $Z_{\rm att} = Z_{\rm att}(0)$. In contrast, for strong pumps, the density first overshoots and subsequently decays to the steady state. This is a consequence of a large occupation number of low-energy polarons building up over time, which enhances momentum relaxation to the light cone through stimulated scattering. Once this happens, radiative loss significantly depletes the population leading to a decrease of the density at late times, see also Appendix D.

The development of a strongly peaked occupation number at the edge of the light cone is clearly visible in the main panel of Fig. 1(c). The initially broad distribution accumulates over time at low momenta, where the relaxation rate $\Gamma_{\rm att}(\mathbf{k}) = (1/V)\sum_{|\mathbf{q}|<|\mathbf{k}|} W_{\mathbf{q}\mathbf{k}}^{\rm att,att}$ scales as $|\mathbf{k}|^3$, see Appendix E. Relaxation, therefore, slows down considerably at low momenta creating a bottleneck. Once they reach the light cone, however, polarons can decay rapidly by creating a photon, hence, suppressing the polaron occupation in the immediate vicinity of $k = 0$. At later times the occupation number at the edge of the light cone grows above one and stimulated scattering leads to the formation of a relatively narrow peak in the steady state, cf. Appendix D.

## IV. STEADY-STATE POLARON DISTRIBUTION

The qualitatively different relaxation dynamics at weak and strong driving also manifests itself in a characteristic steady-state distribution, $n_{\rm att}^\infty(\mathbf{k})$, which we approximate by $n_{\rm att}(\mathbf{k}, t = 10\tau_{\rm exc})$. In Fig. 2(a) we show $n_{\rm att}^\infty$ as a function of $k = |\mathbf{k}|$ for various pump strengths $\Omega$. Our approximation to ignore the time evolution of the electronic bath remains justified up to $\Omega \simeq 10^{-5}E_B q_B^2$, where the electron density exceeds the polaron density by an order of magnitude, and it breaks down for $\Omega = 10^{-4}E_B q_B^2$, where the difference is merely a factor of two. For small $\Omega$ the polaron density is mostly determined by nonradiative loss as only a tiny fraction of polarons enters the light cone, rendering the radiative decay inefficient. Upon increasing $\Omega$, a peak emerges at the edge of the light cone (shaded gray region at the bottom of the figure) and most polarons, being concentrated in this peak, now decay radiatively. A power-law tail $|\mathbf{k}|^{-\alpha}$ emerges next to the peak [see collapse of purple and blue curve with $\alpha \simeq 1.1$ in Fig. 2(a)]. This tail results from stimulated scattering directly to the peak and is reminiscent of, but different from the power-laws characterizing turbulent cascades [50].

Indeed, the relaxation of the states at high momentum $k_{\rm peak} \ll |\mathbf{k}| \lesssim q_B$ is dominated by stimulated scattering into the low-momentum peak with the rate

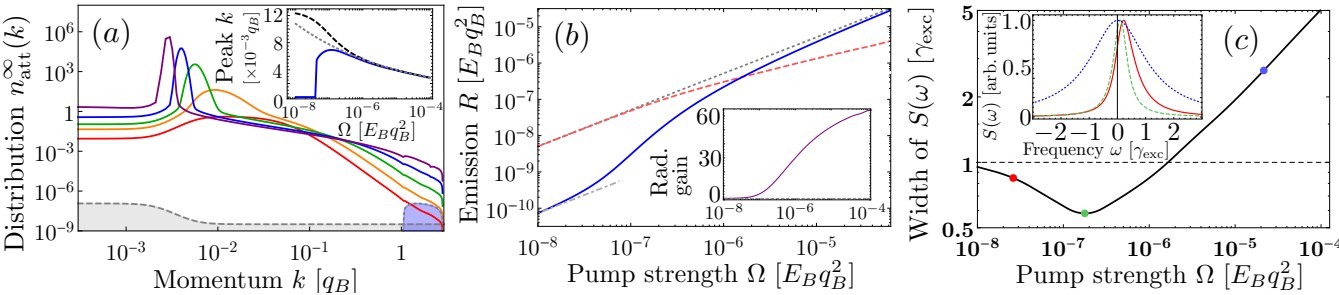

FIG. 2. (a) Late-time distribution function $n_{\text{att}}^{\infty} = n_{\text{att}}(k, t = 10\tau_{\text{exc}})$ in log-log scale for various pump strengths: $\Omega = 10^{-8}, 10^{-7}, 10^{-6}, 10^{-5}, 10^{-4}\ E_B q_B^2$ (from bottom to top at $k = 0$). The loss profile $\gamma(\mathbf{k})$ (gray shaded region) and the effective pump profile $P_{\text{att}}$ for attractive polarons (blue shaded region on the right) are displayed at the bottom (shifted vertically and not to scale). Inset: Momentum-space peak position of the polaron distribution (black dashed line) and emitted light distribution (blue solid line) as a function of $\Omega$ along with the power-law $\Omega^{-1/7}$ (grey dotted line). (b) Pump strength dependence of photon emission rate (solid blue), total polaron decay rate (dotted gray line), and nonradiative loss rate $R_{\text{nrad}}$ (red dashed line) along with the linear scaling at small $\Omega$ corresponding to a radiative efficiency $\eta_{\text{rad}} = 0.73\%$ (dotted-dashed gray line). Inset: radiative gain (for small pump strengths equal to 1). (c) Main panel: Full width at half maximum of the photon spectral function $S(\omega)$. Inset: Normalized spectral function $S(\omega)$ for various pump strengths indicated by the dots in the main panel corresponding to (from left to right) the solid, dashed and dotted curves [frequencies are measured from $\varepsilon_{\text{att}}(0)$].

$C_{\text{out}}(\mathbf{k}) \propto n_{\text{att}}^{\infty}(k_{\text{peak}}) W_{k_{\text{peak}},k}^{\text{att,att}}$, where due to the cylindrical symmetry $W_{kk'} \propto \int d\varphi W_{\mathbf{k},\mathbf{k}'}^{\text{att,att}}$ and $\varphi$ is the angle between $\mathbf{k}$ and $\mathbf{k}'$. Assuming that in the considered region of momenta the effective pump coming from the decay of repulsive polarons and the loss of polarons can be neglected, in the steady state the distribution satisfies a simple rate equation

$$0 \equiv \partial_t n_{\text{att}}^{\infty}(\mathbf{k}) = -C_{\text{out}}(\mathbf{k}) n_{\text{att}}^{\infty}(\mathbf{k}) + C_{\text{in}}(\mathbf{k})[n_{\text{att}}^{\infty}(\mathbf{k}) + 1], \tag{7}$$

where the incoming rate is given by $C_{\text{in}}(\mathbf{k}) = (1/V) \sum_{\mathbf{k}'} W_{\mathbf{k},\mathbf{k}'}^{\text{att,att}} n_{\text{att}}^{\infty}(\mathbf{k}')$. Solving for the distribution function, we find in the selected momentum range

$$n_{\text{att}}^{\infty}(\mathbf{k}) \approx \frac{1}{C_{\text{out}}(\mathbf{k})/C_{\text{in}}(\mathbf{k}) - 1}. \tag{8}$$

Assuming that the incoming rate $C_{\text{in}}(\mathbf{k})$ only weakly depends on momentum, which is confirmed by evaluating it numerically, we replace it by a constant. Therefore, we find

$$n_{\text{att}}^{\infty}(\mathbf{k}) \approx \frac{1}{\beta W_{k_{\text{peak}},|\mathbf{k}|} - 1}, \tag{9}$$

where $\beta$ is a constant. This form of the distribution function matches the numerical solution of the Boltzmann equation very well, and we find that the latter is approximated with a power-law tail $|\mathbf{k}|^{-\alpha}$, where the exponent slightly exceeds 1.

By further increasing the pump strength, the peak position $k_{\text{peak}}$ moves deeper into the light cone. Polarons leave the peak mostly through radiative decay so that $n_{\text{att}}^{\infty}(k_{\text{peak}}) \propto \Omega/\gamma_{\text{rad}}(k_{\text{peak}})$. A small fraction of polarons is scattered with rate $\Gamma_{\text{att}}(k_{\text{peak}}) n_{\text{att}}^{\infty}(k_{\text{peak}})$ to even lower momenta in the interior of the light cone where the occupation $n^*$ is below one. Those states subsequently decay radiatively with rate $Z_{\text{att}} \gamma_{\text{rad}}(k)$, which

we approximate by a constant $\gamma_0$ to gain analytical insight, i.e, we have $n^* \propto \Gamma_{\text{att}}(k_{\text{peak}}) n_{\text{att}}^{\infty}(k_{\text{peak}})/\gamma_0$. The scaling of the peak position with power can be then determined from the condition $n^* \sim 1$, which yields $\Omega \propto \gamma_0 \gamma_{\text{rad}}(k_{\text{peak}})/\Gamma_{\text{att}}(k_{\text{peak}})$ and a scaling

$$k_{\text{peak}} \propto \Omega^{-1/7} \tag{10}$$

in the tail of the radiative loss profile. Indeed, the numerical peak position shown as a black dashed line in the inset of Fig. 2(a) very accurately obeys this power law (dotted gray line) as a function of $\Omega$. Deviations at small power originate from nonradiative processes. Hence, the peak slowly penetrates the light cone further with increasing $\Omega$, which has important consequences for the properties of the emitted light discussed below.

## V. LIGHT EMISSION

The formation of the peak in the distribution function at the edge of the light cone is a generic feature of our driven-dissipative system at strong driving. It results from the competition of radiative decay, which is enhanced at low momenta by light-matter interaction, and polaron relaxation, which is weak at low momenta reflecting the small phase space volume available for scattering. This is in contrast to standard exciton-polariton condensates, where the bottleneck effect occurs due to a significant reduction of the density of states in the strong coupling limit and the population of the $\mathbf{k} = 0$ mode results from exciton-exciton interactions [48].

An important experimental observable is the in-plane momentum of the emitted light, which is peaked around $k = 0$ at weak power. At strong power, the accumulation of polarons at the edge of the light cone instead results in an intensity peak at nonzero momentum, providing clear

evidence for stimulated scattering. The plot of the peak position as a function of power in the inset of Fig. 2(a) shows a jump from zero to a finite value at a threshold $\Omega_{\rm th} \simeq 5 \times 10^{-8} E_B q_B^2$ and a subsequent decay tracing the peak of the polaron distribution $k_{\rm peak}$.

The unusual shape of the distribution has consequences for the emitted radiation, quantified by the emission rate per unit area in the steady state $R(\Omega) \equiv (1/V) \sum_{\bf k} \gamma_{\rm rad}({\bf k}) n_{\rm att}^\infty({\bf k})$. The emitted light intensity initially scales linearly with pump power [see Fig. 2(b)] with a small radiative efficiency $\eta_{\rm rad} = R/\Omega \simeq 0.7\%$, as only a small fraction of polarons are within the light cone. At pump strengths above the threshold $\Omega_{\rm th}$, emission strongly increases and the radiative efficiency approaches unity at high powers. This growth is accompanied by a decreasing total nonradiative decay rate $R_{\rm nrad} \equiv (1/V) \sum_{\bf k} \gamma_{\rm exc} n_{\rm att}^\infty({\bf k})$ (see the dashed red line), while the total decay rate $R + R_{\rm nrad}$ retains the linear scaling with pump strength. The radiative gain, defined as the ratio of the radiative efficiencies at $\Omega$ and at $\Omega \to 0$ [see inset of Fig. 2(b)], equals 1 for weak pumps and increases sharply beyond the threshold. It features an inflection point at a larger pump strength, where the population peak reaches the flat part of the radiative loss profile. For a typical optical transition in TMDs at $\omega \approx 1.6$ eV, we obtain a pumping power density $P_{\rm th} = \omega \Omega_{\rm th} \approx 8.1$ W cm$^{-2}$. This threshold is only an order of magnitude larger than the recently reported laser based on a TMD monolayer nanocavity with ultralow threshold [40].

## VI. SPATIOTEMPORAL COHERENCE

A distinct peak in the distribution function indicates increased spatiotemporal coherence. Here, we focus first on the temporal coherence and discuss the spatial counterpart below. The adiabatic relation between the photon and the exciton explained above permits us to express the photon spectrum near $\omega = \varepsilon_{\rm att}(0)$ as $S(\omega) \simeq (g^2/V) \sum_{\bf k} n_{\rm att}({\bf k}) \mathcal{A}_{\rm att}({\bf k}, \omega)/[({\bf k}^2/2m_{\rm ph})^2 + \gamma_{\rm ph}^2/4]$ in terms of the attractive polaron spectral function $\mathcal{A}_{\rm att}({\bf k}, \omega)$ displayed in Fig. 1(b). For a weak drive, the spectrum, shown as a solid red line in the inset of Fig. 2(c), has an asymmetric lineshape with a high-frequency tail as a result of the relatively broad momentum distribution of polarons. The spectral peak changes nonmonotonously as a function of pump power, being narrowest for intermediate powers (green dashed line).

The peak width plotted in the main panel of Fig. 2(c) initially decreases as a function of power until $\Omega = 10^{-7} E_B q_B^2$ as the polaron distribution develops a low-energy peak [cf. the orange curve in Fig. 2(a)]. Beyond this point, the linewidth rapidly increases as the radiative loss becomes more prominent. Interestingly, the minimal width is considerably smaller than the bare exciton linewidth $\gamma_{\rm exc}$, which is possible because of the reduced quasiparticle weight $Z_{\rm att} < 1$. That is, the composite

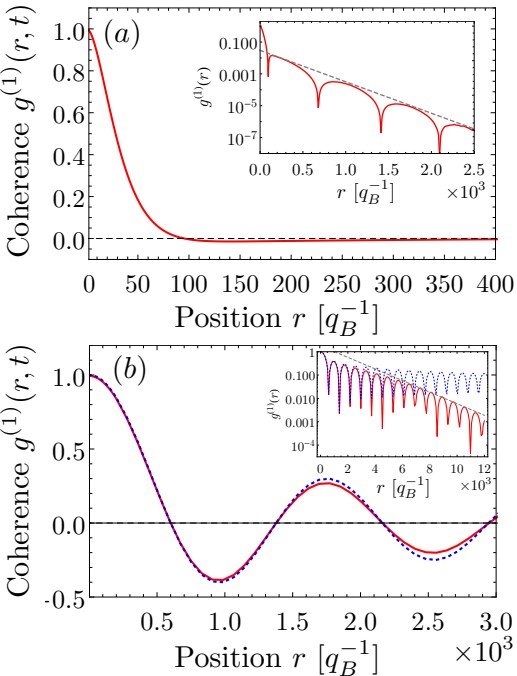

FIG. 3. The coherence function $g^{(1)}(r, t)$ (solid red line) for $t = 10\tau_{\rm exc}$ and for weak and strong pumps, i.e., $\Omega/E_B q_B^2 = 10^{-8}$ (a) and $10^{-4}$ (b). (Insets) The long scale behaviour of $g^{(1)}(r, t)$ (notice the difference in the horizontal scales). The dashed gray lines — exponential tails $\sim e^{-r/\xi}$, with $\xi = 2.2 \times 10^2 q_B^{-1}$ (a) and $1.8 \times 10^3 q_B^{-1}$ (b). The dotted blue line in the lower panel — Bessel functions $J_0(k_{\rm peak} r)$.

nature of exciton polarons allows for a narrowing of the laser linewidth below the limit for bare excitons over a range of powers up to $\Omega \simeq 2 \times 10^{-6} E_B q_B^2$, where the radiative emission is characterized by a relatively large gain $\approx 40$ and the linewidth $\gamma_{\rm exc}$ is 0.15% of the bare photon linewidth $\gamma_{\rm ph}$. The minimal linewidth can be even further reduced by reducing the density of electrons (thereby reducing $Z_{\rm att}$) at the cost of increasing the formation time of the coherent polaron peak.

The particular change in the form of the nonequilibrium distribution function of the attractive polarons below and above the threshold has important implications for the spatial coherence function of the system. To characterize how it changes across the threshold in pump strength, we refer to the coherence function $g^{(1)}(r, t) = G^{(1)}(r, t)/|G^{(1)}(r, t)|$, where $G^{(1)}(r, t)$ is the one-body correlation function given by the Fourier transform of $\langle \hat{x}_{\bf k}^\dagger(t) \hat{x}_{\bf k}(t) \rangle$. The contribution to the coherence comes from the attractive polarons, as the occupation of the repulsive polarons is orders of magnitude smaller. After projecting on the energy shell, the correlation function is given by $G_{\rm att}^{(1)}(r, t) = \int_0^\infty \frac{k dk}{2\pi} J_0(kr) Z_{\rm att}(k) n_{\rm att}(k, t)$.

Fig. 3a shows the coherence function $g^{(1)}(r, 10\tau_{\rm exc})$ for a weak pump strength below threshold $\Omega = 10^{-8} E_B q_B^2$. The coherence drops sharply on a scale $\sim 30 q_B^{-1}$, which

for physical parameters is 0.074 $\mu$m. After this sharp drop, $g^{(1)}$ falls off exponentially as $\sim e^{-r/\xi}$ with $\xi = 220 q_B^{-1}$ which corresponds to 0.54 $\mu$m (see the dashed gray line in the inset). The rapid drop on the shortest scale is related to a relatively broad distribution of the polarons in momentum space.

In Fig. 3b, we plot the coherence function (solid red) for a pump strength above the threshold $\Omega = 10^{-4} E_B q_B^2$. Here the emergence of the peak in the distribution manifests itself in the presence of the much slower drop of the coherence function up to relatively far distances. For $r \lesssim 2/\Delta k \approx 3.7 \times 10^3 q_B^{-1}$ (corresponding to 9.1 $\mu$m), where $\Delta k$ is the width of the peak in $n_{\mathrm{att}}$, the coherence is dominantly described by a single Bessel function $J_0(k_{\mathrm{peak}} r)$, see the dotted blue line. On the other hand, the second regime $r \gtrsim 2/\Delta k$ is characterized by an exponential drop $e^{-r/\xi}$, with $\xi = 1.8 \times 10^3 q_B^{-1}$.

## VII.   CONCLUSIONS

We have described a lasing transition for exciton polarons weakly coupled to photons, based on a kinetic equation derived from non-equilibrium quantum-field theory. The intricate relaxation dynamics of Fermi polarons under drive and dissipation result in a non-monotonous power dependence of the laser linewidth. Besides being relevant for the modeling of semiconductor light sources, the richness of the underlying non-equilibrium dynamics sheds light on the quantum many-body nature of Fermi polarons, both in solid-state materials [33] and ultracold atomic gases [51–54]. Finally, observing signatures of stimulated emission, for instance, from the momentum dependence of the emitted light, could serve as a unique signature of the bosonic nature of exciton polarons distinguishing them from fermionic trions [18]. An interesting future direction is to include scattering from phonons or disorder, which is not expected to qualitatively change the relaxation dynamics associated with the bottleneck effect, but could alter the quantitative power dependence. Moreover, a full description of the electron-mediated interaction will require to properly include the dynamics of the Fermi-surface.

## ACKNOWLEDGMENTS

We acknowledge helpful discussions with Kristiaan de Greve and Andrey Sushko. F. Pientka was supported by the Deutsche Forschungsgemeinschaft (DFG, German Research Foundation) through TRR 288 - 422213477 (project B09).

## Appendix A: Derivation of the kinetic equation

In this section we derive kinetic equations, based on the formalism of non-equilibrium quantum field theory within Keldysh approach [55, 56], describing the transitions between the polarons resulting from collisions with electrons from a bath.

The Hamiltonian that describes excitons, electrons, and the interaction is in Eq. (1) in the main text, but for convenience we repeat it here:

$$\hat{H} = \hat{H}_x + \hat{H}_e + \hat{H}_{\mathrm{int}}. \tag{A1}$$

The free Hamiltonians are: $\hat{H}_x = \sum_{\mathbf{k}} \varepsilon_x(\mathbf{k}) \hat{x}_{\mathbf{k}}^\dagger \hat{x}_{\mathbf{k}}$ and $\hat{H}_e = \sum_{\mathbf{k}} \varepsilon_e(\mathbf{k}) \hat{e}_{\mathbf{k}}^\dagger \hat{e}_{\mathbf{k}}$ with kinetic energies $\varepsilon_x(\mathbf{k}) = \mathbf{k}^2/2m_x$ and $\varepsilon_e(\mathbf{k}) = \mathbf{k}^2/2m_e - E_F$ of excitons and electrons, respectively. We denote with $\mathbf{k}$ an in-plane momentum of the particles and throughout the text we set $\hbar = 1$. The operators $\hat{x}_{\mathbf{k}}$ ($\hat{e}_{\mathbf{k}}$) are bosonic (fermionic) annihilation operators of excitons (electrons). We assume that the inter-species interaction is a contact potential with strength $U$, i.e., $\hat{H}_{\mathrm{int}} = U \int d^2 r\, \hat{x}^\dagger(\mathbf{r}) \hat{x}(\mathbf{r}) \hat{e}^\dagger(\mathbf{r}) \hat{e}(\mathbf{r})$. This form of interaction has been used in the literature to describe the Fermi polaron problem [18, 27].

To model the action of loss and external drive of excitons we employ the quantum master equation [56]

$$\partial_t \hat{\varrho}(t) = -i[\hat{H}, \hat{\varrho}(t)] + \mathcal{L}_d \hat{\varrho}(t). \tag{A2}$$

The operator $\mathcal{L}_d$ is a sum of two terms [37]. The first part, given by $\sum_{\mathbf{k}} \gamma(\mathbf{k}) D[\hat{x}_{\mathbf{k}}]$, where $D[\hat{x}_{\mathbf{k}}]\hat{\varrho} \equiv \hat{x}_{\mathbf{k}}^\dagger \hat{\varrho} \hat{x}_{\mathbf{k}} - \frac{1}{2}\{\hat{x}_{\mathbf{k}}^\dagger \hat{x}_{\mathbf{k}}, \hat{\varrho}\}$, describes the loss channel with a rate $\gamma(\mathbf{k})$ of excitons moving with momentum $\mathbf{k}$.

The second term in the operator $\mathcal{L}_d$ is given by $\sum_{\mathbf{k}} \Omega(\mathbf{k}) P[\hat{x}_{\mathbf{k}}]$, where the pump operator is $P[\hat{x}_{\mathbf{k}}] \equiv D[\hat{x}_{\mathbf{k}}] + D[\hat{x}_{\mathbf{k}}^\dagger]$, and it describes reinjection of excitons with a rate $\Omega(\mathbf{k})$ [56–58]. Although in our starting equation, as seen from Eq. (A2), the pump is time and frequency independent, we depart from this assumption after upgrading the formalism to Keldysh path-integrals [55, 56].

### 1.   Non-equilibrium QFT

At this point, as we are interested in non-equilibrium description of the system, we resort to Keldysh description. Namely, instead of working directly with Eq. (A2), we rephrase the problem in terms of path-integral generating functional $\mathcal{Z}$ expressed in terms of the Keldysh action $S$, for details see Refs. [55, 56]. The unit-normalized functional integral $\mathcal{Z}$ explicitly reads

$$\mathcal{Z} = \int \mathcal{D}\phi\, \mathcal{D}\psi\, e^{iS[\bar{\phi},\phi,\bar{\psi},\psi]}, \tag{A3}$$

where the integration measure $\mathcal{D}\phi \equiv \prod_{\alpha=c,q} \mathcal{D}\bar{\phi}^\alpha \mathcal{D}\phi^\alpha$ is over classical $\phi^c(x)$ and quantum $\phi^q(x)$ components in the bosonic Keldysh space (K) of the complex bosonic exciton field $\phi = (\phi^c, \phi^q)^T$ as well as their conjugate fields; here $x = (\mathbf{r}, t)$ and $^T$ stands for matrix transposition. The electron field $\psi(x) = (\psi_1(x), \psi_2(x))^T$ is a vector (in

fermionic Keldysh space) field of anticommuting Grassmann variables, and by $\bar{\psi} = (\bar{\psi}_1, \bar{\psi}_2)^T$ we denote the conjugate field; similarly to the bosonic case, the integration measure is $\mathcal{D}\psi \equiv \prod_{a=1,2} \mathcal{D}\bar{\psi}_a \mathcal{D}\psi_a$.

The formalism that we employ in this work closely follows the one developed in Ref. [37]. In short, the Keldysh action consist of three terms corresponding to excitons, electrons and their interaction, i.e.,

$$S = S_x + S_e + S_{\text{int}}, \tag{A4}$$

where the free actions are:

$$S_x = \int dx\, dx'\, \bar{\phi}^\alpha(x) D_x^{\alpha\beta}(x,x') \phi^\beta(x'), \tag{A5a}$$

$$S_e = \int dx\, dx'\, \bar{\psi}_a(x) D_e^{ab}(x,x') \psi_b(x') \tag{A5b}$$

and with $dx = d^2 r dt$ (the summation over repeated indices in implied). The formulas listing the *bare* Green functions $\hat{G}_{0,i} \equiv \hat{D}_i^{-1}$, $i = x, e$ are presented in Sec. B; by the hat symbol we denote $2 \times 2$ matrices acting in the Keldysh space.

The action $S_{\text{int}}$ in the path integral picture consists of local terms that schematically are $\sim \bar{\phi}\phi\bar{\psi}\psi$. At small exciton densities the trion state, i.e., the molecular state of an exciton and an electron, is strongly coupled to bare exciton states and as a result repulsive and attractive polaron branches emerge in the excitation spectra [27, 33, 37]. To take into account nonperturbatively the electron-exciton pairing [37, 59–61], with the help of the Habbard-Stratonovich transform we decouple the action in the pairing channel at a cost of introducing an auxiliary fermionic field $\Delta \sim \phi\psi$ that carries a Keldysh index and is described by the bare action

$$S_\Delta = \int dx\, \bar{\Delta}_a D_\Delta^{ab} \Delta_b, \tag{A6}$$

that should be added to the total action $S$ while the functional $\mathcal{Z}$ should be supplemented with an additional integration $\int \mathcal{D}\Delta$. As a result the interaction now contains the terms $\bar{\phi}\phi\bar{\psi}\psi \sim \bar{\Delta}\psi\phi + \bar{\psi}\bar{\phi}\Delta$. Such a form describes the process of annihilation of an exciton and electron and creation of a molecule accompanied with a reverse process.

### 2. Dyson equation

Due to the interaction between the species via the interaction mediating field $\Delta$, the propagation of the particles is modified manifesting in the Dyson equation for the particle propagator [55], i.e., the inverse of the dressed GFs:

$$\hat{G}_i^{-1} = \hat{D}_i - \hat{\Sigma}_i \tag{A7}$$

for $i = x, e, \Delta$, in which the self-energies $\hat{\Sigma}_i$ quantify the impact of the interactions on the propagators. In

Sec. C, we provide the details for the evaluation of the self-energies $\hat{\Sigma}$ to one-loop order [62] in the conserving approximation [63–67]. They read:

$$\Sigma_\Delta^{aa'}(x,x') = \frac{i}{2} G_x^{\alpha\alpha'}(x,x') \big[ \hat{\gamma}^\alpha \hat{G}_e(x,x') \hat{\gamma}^{\alpha'} \big]_{aa'}, \tag{A8a}$$

$$\Sigma_x^{\alpha\alpha'}(x,x') = -\frac{i}{2} \text{Tr}[\hat{\gamma}^\alpha \hat{G}_\Delta(x,x') \hat{\gamma}^\alpha \hat{G}_e(x',x)], \tag{A8b}$$

$$\Sigma_e^{aa'}(x,x') = \frac{i}{2} G_x^{\alpha\alpha'}(x',x) \big[ \hat{\gamma}^{\alpha'} \hat{G}_\Delta(x,x') \hat{\gamma}^\alpha \big]_{aa'}, \tag{A8c}$$

where the matrices $\gamma_{ab}^c = \delta_{ab}$ and $\gamma_{ab}^q = 1 - \delta_{ab}$ act in the Keldysh fermionic subspace, and on the right-hand sides the dressed GFs are used; the trace acts in the Keldysh space of Keldysh indices. Notice the reverse order of arguments in exciton and electron functions in the second line. This is physically related to a virtual creation of a molecule when a propagating exciton collides with an electron. The molecular self-energy describes a decay of the trion into an electron and exciton and so the same order of arguments in $\hat{G}_e$ and $\hat{G}_x$ in the first line.

The Dyson equation contains information about the spectrum of excitations of the system as well as the quantum kinetic equation. We first extract the spectrum, i.e., the repulsive and attractive polaron branches, and then proceed to the description of their non-equilibrium relaxation to a stationary state under continuous external drive.

### 3. Retarded GF and distribution function

To unravel the separation of GFs into distribution and spectral functions we take advantage of the Keldysh structure of the GFs, see Sec. B and Ref. [55] for details. Thus, the components $G_x^{cq}(x,x')$, $G_{\Delta/e}^{11}$ and $G_x^{cc}$, $G_{\Delta/e}^{12}$ yield, respectively,

$$G_i^R = (D_i^R - \Sigma_i^R)^{-1}, \tag{A9a}$$

$$G_i^K = G_i^R \circ (-D_i^K + \Sigma_i^K) \circ G_i^A, \tag{A9b}$$

where the convolution symbol stands for matrix multiplication in the spacetime domain; the inverse in the first line is taken with respect to this multiplication. The retarded (R), advanced (A) and Keldysh (K) component are the elements of the corresponding matrices that highlights the retarded (advanced) property, i.e., $G^{R(A)}(x,x') = 0$ if $t < t'$ ($t > t'$). While the retarded GFs provides the spectrum of excitations, the Keldysh GF gives access to the distribution function $F$ of excitations by $G_i^K = G_i^R \circ F_i - F_i \circ G_i^A$ [55]. It is useful to extract the part $\delta F_i$ of $F_i$ that is proportional to the occupation of particles by $F_i = 1 + \delta F_i$, and denote the corresponding part of $G_i^K$ by $\delta G_i^K$.

### 4. Approximations

Now we discuss the two main approximations involved in our theory. First, the coupling of the electrons with

excitons leads to a bound trion state that is described by a resonance at $E_B$ redshifted from the continuum threshold in the spectral function of the molecules even in the limit of vanishing electron density. Since we are interested in the impurity limit when the density of excitons is much smaller than the density of electrons, in the first order the molecular spectral function will be modified do to the presence of the free fermionic carriers and the Pauli blocking [27, 37]. We therefore neglect the contribution in $\Sigma_\Delta^R$ coming from the term $\delta G_x^K$. Consequently, we neglect in $\Sigma_x^R$ the contribution from $\delta G_\Delta^K$, which is also proportional to the density of impurities $\delta G_x^K$. In this way, the spectral functions are tuned in the first approximation by the density of electrons and are independent of the exciton density.

The second approximation concerns the state of electrons. Due to coupling to the lattice and by the diffusion of the heat through the boundaries of the sample the electron gas reaches thermal equilibrium on a fast timescale [68]. We therefore assume that electrons are always in thermal equilibrium and neglect small deviations around the Fermi surface. This neglected effect is related to phase-space filling (PSF) effect which leads to residual interactions between polarons and induces a shift of the polaron resonances [33] that is much smaller than the separation between the polarons. The study of the impact of PSF effect on dynamics is included in our theory, but it is beyond the scope of this work and will be the subject of research in the future. Consequently, we assume that $\Sigma_e^R \approx 0$ and $G_e^K$ is set by the Fourier transform of $F_e(x, x')$ that is given by the fluctuation dissipation relation (FDR) [55], i.e., $\delta F_e(k) = -2n_e(\omega)$, where we denote the momentum-frequency vector $k = (\mathbf{k}, \omega)$ and $n_e(\omega)$ is the Fermi-Dirac distribution at $T = 0$, i.e., $n_e(\omega) = \theta(\epsilon_F - \omega)$ parameterized by Fermi energy $E_F$.

### 5. Spectral function

Basing on the approximations, which should be valid in the impurity limit $\rho_x \ll n_e$, where $\rho_x$ ($n_e$) is the density of excitons (electrons), the spectral functions do not depend on the density of excitons, and we can proceed to the calculation of the self-consistent self-energies and Green functions. That is, we calculate in the steady state the GFs: $G_x^R(k) = 1/(D_x^R(k) - \Sigma_x^R(k))$ and $G_\Delta^R(k) = 1/(D_\Delta^R(k) - \Sigma_\Delta^R(k))$, where the self-energies depend also on $G_x^R(k)$ and $G_\Delta^R(k)$. Here, we take the case $m_x = 2m_e$ and $E_F = E_B$; we also set the unit of wave vectors equal to $E_B = q_B^2/m_e$.

In Fig. 1(b) in the main text we present the results of the self-consistent calculations for this parameters of the excitation spectrum in the system. Since it is computationally hard to resolve in momenta the light cone ($k \ll q_B$) and in frequencies the features corresponding to the lifetime $\tau_{\text{exc}} \sim 1$ ns, we resorted to a simplification. Namely, we assumed $\gamma(\mathbf{k})$ is constant in momentum and of the order of a percent of $E_B$ (much smaller than the

width in of the resonances in Fig. 1(b) in the main text. We expect that the position of the resonances as well as the spectral weights associated with resonances are not much influenced by inclusion of the light cone. Similarly, the widths of the resonances in the spectral functions are not important for our theory presented below, as the lifetime of the polarons is captured by the kinetic equation. Consequently, in the kinetic equation the precise form of the loss profile $\gamma(\mathbf{k})$ has an important consequences for the distribution function and is kept in the initial form. In our numerical calculations, we discretize the $|\mathbf{k}|$ and $\omega$ space with a large cutoff $\Lambda$. We find $G_\Delta^R$, and for $n_e = 0$ we fix the position of the position of the bound state $E_B$ by tuning the exciton-electron interaction strength $U$.

### 6. Polaron resonances

In Fig. 1(b) we show the spectral function $\mathcal{A}_x(k, \omega) = -2\text{Im}[G_x^R(k, \omega)]$. We find the maximum of the resonances in the upper and lower polaron branches, which yields the polaron dispersion relation, i.e., $\varepsilon_\alpha(\mathbf{k})$, where $\alpha = $ rep or att for repulsive (higher in energy) and attractive (lower in energy) polarons, respectively. These function are shown with dashed lines in Fig. 1(b). The attractive polaron ceases to be a sharp resonance for momenta $k \gtrsim q_B$ when it enters the trion-hole continuum. In Fig. 4a we show the coupling matrix $|T(Q, \omega)|^2$, defined by the relation

$$T(\mathbf{Q}, \omega) \equiv G_\Delta^R(\mathbf{Q}, \omega), \qquad (A10)$$

which is important for transitions between polaron states in the kinetic equation. The sharp feature (notice the logarithmic scale) induces a rapid decay from the repulsive polaron into high-momentum states of the attractive polaron.

Finally, we note that the spectral weights associated with both polarons is momentum-dependent. We define $Z_\alpha(\mathbf{k})$ as the integral over the frequencies around the resonances. Specifically, since the polaron resonances are asymmetric, as a natural border between the resonances we take the maximum of the molecular spectral function. Therefore, $Z_{\text{att}}(\mathbf{k})$ ($Z_{\text{rep}}(\mathbf{k})$) results from integration over the frequencies smaller (greater) than the position of the molecular peak for each $\mathbf{k}$. In Fig. 4b, we provide the results, which indicate that in this case, the dependence on wave vector is rather modest.

### 7. Derivation of the kinetic equation

To proceed with the description of the system out of equilibrium, we parameterize $G_i^K$ in terms of the hermitian matrices $F_i$ and rewrite Eq. (A9b) for $i = x$ as $F_x \circ [G_x^A]^{-1} - [G_x^R]^{-1} \circ F_x = -D_x^k + \Sigma_x^K$. In the stationary state without the loss and the pump, this equation leads to FDR, i.e., $\Sigma_x^K = \Sigma_x^R \circ F_x^{\text{th}} - F_x^{\text{th}} \circ \Sigma_x^A$, where

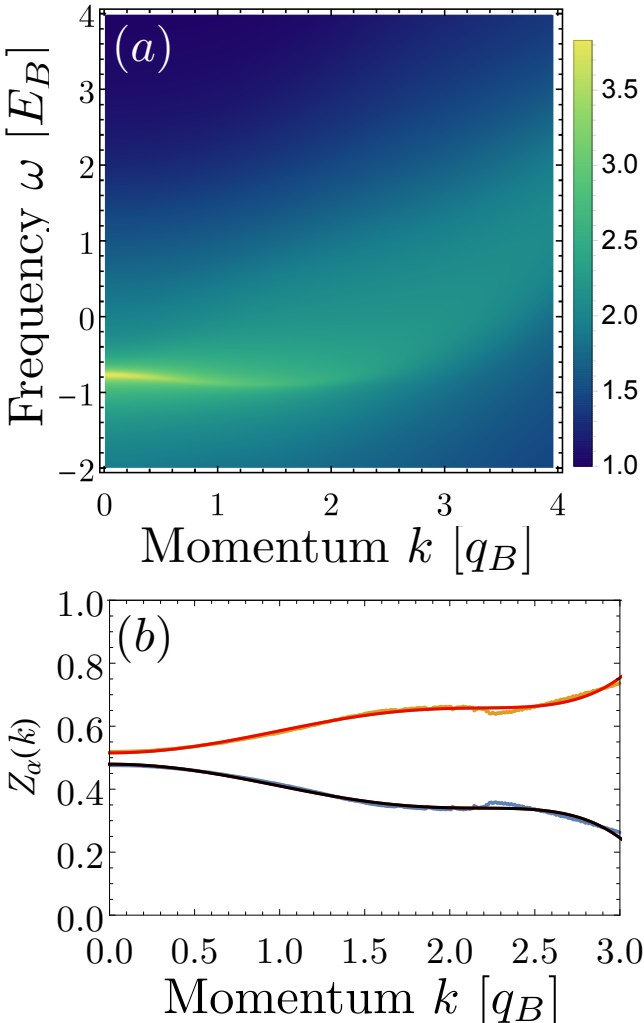

FIG. 4. (a) The coupling matrix $\log_{10}|T|^2$ as a function of $|\mathbf{Q}|$ and $\omega$ for $E_F = E_B$ and $m_x = 2m_e$; $T(\mathbf{Q},\omega)$ is in the units of $E_B q_B^2$. (b) Spectral weights $Z_\alpha(\mathbf{k})$ as a function of $|\mathbf{k}|$. Calculated as the integral of the spectral function $\mathcal{A}_x(\omega,\mathbf{k})$ over the frequencies up/from to the maximum of the molecular peak in the molecular spectral function. To the numerically calculated values, we fit the function $Z_\alpha(k) = Z_0^\alpha(1 + \beta_2^\alpha k^2 + \beta_4^\alpha k^4 + \beta_6^\alpha k^6)$, where $\alpha = $ att and rep (lower and upper lines, respectively).

the thermal distribution function is $F(\omega) = 1 + 2n_x^{\mathrm{th}}(\omega)$, with $n_x^{\mathrm{th}}(\omega)$ being the thermal distribution of bosons, i.e., $1/[e^{(\omega-\mu_x)/T}-1]$. In such a case, the Keldysh component carries no new information and is fully specified by the temperature of the electron bath and chemical potential which sets the total density. In our non-equilibrium case, the drive and dissipation are playing the prominent role and thus the Keldysh component is independent. Since we expect that the relaxation will be dominated by the loss and drive, we provide the results for the electron bath at $T = 0$.

To make further progress we apply the Wigner transform [55, 56] to the equation for $F_x$ and expanding in

the central time up to linear order in gradients we arrive at the quantum kinetic equation for the distribution function $F_x(x,k)$:

$$i\{F_x, \omega - \tilde{\omega}_x\} = i\gamma(\mathbf{k})F_x - D_x^K - i\tilde{I}_x[F_x] \qquad \text{(A11)}$$

where $\tilde{I}_x = i\Sigma_x^K - iF_x(\Sigma_x^R - \Sigma_x^A)$ is the collision integral, $\tilde{\omega}_x(x,k) = \varepsilon_x(\mathbf{k}) + \mathrm{Re}[\Sigma_x^R(x,k)]$ is the renormalized exciton energy, and all the function are now evaluated at $(x,k)$, $x$ being the central spacetime variable and $k$ the energy-momentum corresponding to the relative variable $x - x'$ after applying the Fourier transform. Here, the Poisson bracket is defined as $\{A,B\} \equiv \partial_x A \partial_k B - \partial_x B \partial_k A$, and $\partial_x B \partial_k A \equiv \partial_{\mathbf{r}} A \cdot \partial_{\mathbf{k}} B - \partial_t A \partial_\omega B$. The left-hand side is the drift term of the kinetic equation whereas on the right-hand side there is the decay (first term) and drive (second term). The third term is the collisional integral. We note that the Keldysh component can be taken as $D_x^K(\mathbf{r},t,\mathbf{k},\omega) = i[\gamma(\mathbf{k})+2\Omega(\mathbf{r},t,\mathbf{k},\omega)]$ generalizing it to space-time and frequency-momentum dependent pump. This term cancels a part of the first term leading to $\delta F_x = 2\Omega/\gamma$ in the stationary homogeneous state (drift term is zero) and without collisions (second line neglected) which indicates that the mean occupation of the momentum modes is $\delta F_x(k)/2 = n_x(\mathbf{k}) = \Omega(\mathbf{k})/\gamma(\mathbf{k})$ as it should be.

We note that the self-energies are nonlinear functions of $F_x$ since it enters in $\hat{\Sigma}_x$ in Eq. (A8b) through $\hat{G}_\Delta$, which in turn depends on $\hat{G}_x$ via Eq. (A8a). Now, in the kinetic equation, all the terms has to be retained in order to describe the interaction between the particles.

As a next step, we write the Wigner transformed electron Keldysh GF as $G_e^K(x,k) = F_e(x,k)[G_e^R(x,k) - G_e^A(x,k)]$ and for the interaction mediating field we use $G_\Delta^K(x,k) = G_\Delta^R(x,k)\Sigma_\Delta(x,k)G_\Delta^A(x,k)$, which are valid up to the first order in the gradient expansion. The collisional integral $\tilde{I}_x(k) \equiv i\Sigma_x^K(k) + 2F_x(k)\mathrm{Im}[\Sigma_x^R(k)]$ obtained in this way takes the form

$$\tilde{I}_x[F_x] = \left(\frac{1}{2V}\right)^2 \sum_{q,q'} |G_\Delta^R(q)|^2 \mathcal{A}_e \mathcal{A}'_e \mathcal{A}''_x \times \qquad \text{(A12a)}$$

$$\times \frac{1}{2}\Big\{ -[F_x(k)+1](F_e+1)(F''_x-1)(F'_e-1) \quad \text{(A12b)}$$

$$+(F''_x+1)(F'_e+1)[F_x(k)-1](F_e-1) \Big\}, \quad \text{(A12c)}$$

where the left-hand side is evaluated at $k$, the function without an argument is evaluated at $(q-p)$, with a prime at $q'$, and with a double-prime at $q - q'$; we also suppressed spacetime variables for clarity. In the line (A12a), $|G_\Delta^R|^2$ plays the role of the coupling matrix $|T|^2$ in the collisions between polarons and electrons, and the spectral functions force the energy conservation. The lines (A12b) and (A12c) describe the "in" and "out" processes, respectively, for the energy-momentum $k$.

The thermal solution $F_x(\mathbf{k},\omega) = \coth(\frac{\omega-\mu_x}{2T})$ and $F_e(\mathbf{k},\omega) = \tanh(\frac{\omega-\mu_e}{2T})$ with the chemical potentials $\mu_i$

and temperature $T$ nullifies the collisional integral irrespective of the precise form of $|G_\Delta^R|^2$. On the other hand, in non-equilibrium under external drive and in the presence of loss, the distribution is sensitive to form of the coupling matrix.

The Eq. (A11) together with Eq. (A12) constitute the basis for the description of the quantum dynamics in the impurity limit where density of excitons $n_x \ll n_e$. It must be supplemented by the equation for the retarded GFs: $G_x^R(x,k) = (D_x^R(k,p) - \Sigma_x^R(x,p))^{-1}$ which is valid up to the inclusion of terms linear in gradient expansion.

The kinetic equation describes evolution of a multidimensional function $F_x(\mathbf{r}, t, \mathbf{k}, \omega)$. To simplify the problem, we project $F_x$ on the polaron energy shells. This is valid if the resonances in the spectral function are much narrower than the characteristic change of the function in frequencies [55]. Although for higher momenta the polaron resonances are broad, due to the low occupation of these modes we expect that the description of the dynamics in terms of quasi-particles is at least qualitatively correct in this regime.

For long-lived quasi-particles the energies $\omega = \varepsilon_\alpha(\mathbf{k})$, with $\alpha = \text{att}$ and rep of the attractive and repulsive polarons, respectively, are determined by zeros of the mass function $\mathcal{M}_x = \omega - \tilde{\omega}_x$. We project Eq. (A11) by multiplying its both sides by $\delta(\mathcal{M}_x)$ and integrating over $\omega$ around the two distinct solutions. Since on the right-hand side a Poisson bracket appears in the form of $\{F_x, \mathcal{M}_x\}$ the projection is particularly simple to evaluate. Now we employ the relation $\partial_u\tilde{\omega}_x|_{\varepsilon_\alpha} = Z_\alpha^{-1}\partial_u\varepsilon_\alpha$, which is valid for projection of the derivatives with $u = \mathbf{k}, \mathbf{r}, t$, and where the inverse of the quasi-particle weight is $Z_\alpha^{-1} = (1 - \partial_\omega\tilde{\omega})|_{\omega=\varepsilon}$. Thus, the left hand side (multiplied with $i$) takes the form

$$\partial_t n_\alpha - \{\varepsilon_\alpha, n_\alpha\} = -\gamma_\alpha(\mathbf{k})n_\alpha + \Omega_\alpha + I_\alpha, \qquad (A13)$$

where the polaron distribution function is $n_\alpha(\mathbf{r}, \mathbf{k}, t) = \delta F_x(\mathbf{r}, t, \mathbf{k}, \omega = \varepsilon_\alpha(\mathbf{k}))/2|_{\omega=\varepsilon_\alpha}$, the Poisson bracket reduces to its classical form with derivatives only over space-momentum variables. From now on we will omit the space variable. Here, the renormalized decay rate is $\gamma_\alpha(\mathbf{k}) = Z_\alpha(\mathbf{k})\gamma(\mathbf{k})$ and the renormalized pump strength is $\Omega_\alpha(\mathbf{k}, t) = Z_\alpha(\mathbf{k})\Omega(t, \mathbf{k}, \omega = \varepsilon_\alpha)$. The collisional integral is given by $I_\alpha = \frac{1}{2}Z_\alpha\tilde{I}_x|_{\omega=\varepsilon_\alpha(\mathbf{k})}$, and explicitly takes the form:

$$I_\alpha = \frac{1}{V}\sum_{\beta=\text{att,rep}}\sum_{\mathbf{k}'}W_{\mathbf{k}\mathbf{k}'}^{\alpha\beta}[n_\alpha(\mathbf{k})+1]n_\beta(\mathbf{k}') \qquad (A14a)$$

$$-\frac{1}{V}\sum_{\beta=\text{att,rep}}\sum_{\mathbf{k}'}W_{\mathbf{k}'\mathbf{k}}^{\beta\alpha}[n_\beta(\mathbf{k}')+1]n_\alpha(\mathbf{k}), \quad (A14b)$$

and the transition rates are given by:

$$W_{\mathbf{k}\mathbf{k}'}^{\alpha\beta} = \frac{2\pi}{V}\sum_{\mathbf{Q}}|G_\Delta^R(\mathbf{Q}, \varepsilon_\beta(\mathbf{k}')+\varepsilon_e(\mathbf{q}'))|^2 \times \quad (A15a)$$

$$\times Z_\alpha(\mathbf{k})Z_\beta(\mathbf{k}')n_e(\mathbf{q}')[1 - n_e(\mathbf{q})] \times \qquad (A15b)$$

$$\times \delta(\varepsilon_\alpha(\mathbf{k})+\varepsilon_e(\mathbf{q})-\varepsilon_e(\mathbf{q}')-\varepsilon_\beta(\mathbf{k}')). \qquad (A15c)$$

Here, $n_e(\mathbf{q})$ is the Fermi distribution at $T = 0$, i.e., $\theta(k_F - |\mathbf{q}|)$, and the electron momenta are: $\mathbf{q}' = \mathbf{Q} - \mathbf{k}'$ and $\mathbf{q} = \mathbf{Q} - \mathbf{k}$. The transition rate describes the transition between polarons as a result of the collision with electrons, schematically:

$$(\beta, \mathbf{k}') + (e, \mathbf{q}') \longrightarrow (\alpha, \mathbf{k}) + (e, \mathbf{q}), \qquad (A16)$$

for which the electron with $|\mathbf{q}'| < k_F$ is scattered outside the Fermi sea, $|\mathbf{q}| > k_F$, and during the collision the total energy and momentum are conserved. Finally, we remark that in thermal equilibrium the solutions are given by $F_e(\mathbf{k}) = \tanh[(\varepsilon_e(\mathbf{k}) - \mu_e)/2T]$ and $F_\alpha(\mathbf{k}) = \coth[(\varepsilon_\alpha(\mathbf{k}) - \mu_x)/2T]$, where $\alpha = \text{att, rep}$ and the chemical potential of polarons is the same for both branches.

## Appendix B: Green functions in Keldysh QFT

The details of transition from operator language to path-integral formulation can be found in Ref. [55] for closed systems and in Ref. [56] for open systems. Hereafter, by $D = G_0^{-1}$ we denote the inverse of the *bare* GFs, and by $G$ the *dressed* GFs; the vectors $k = (\mathbf{k}, \omega)$ and $x = (\mathbf{r}, t)$. The evolution equation, given in Eq. (A2), corresponds to the action $S = S_x + S_e + S_{\text{int}}$. Below, we write down the respective parts.

*Exciton Green functions.* The *bare* exciton action, see also Ref. [37], is given by

$$S_x = \iint dx dx' \bar{\phi}^\alpha(x)D_x^{\alpha\beta}(x,x')\phi^\beta(x'), \qquad (B1)$$

where the inverse bare Green function $\hat{D}$ is related to the bare GF by taking the matrix inverse, i.e., $\hat{G}_{0,x} = \hat{D}_x^{-1}$, and it has a standard causality structure for bosons:

$$\hat{D}_x = \begin{pmatrix} 0 & D_x^A \\ D_x^R & D_x^K \end{pmatrix}. \qquad (B2)$$

Due to the diagonal structure in energy-momentum representation, i.e., after taking the Fourier transform, the entries are most conveniently represented in frequency-momentum space, i.e., the diagonal parts of the GFs are: retarded/advanced $D_x^{R/A}(k) = \omega - \varepsilon_x(\mathbf{k}) \pm i\frac{\gamma(\mathbf{k})}{2}$, and the Keldysh component is $D_x^K(k) = i[\gamma(\mathbf{k}) + 2\Omega(k)]$, where we indicated that the pump may depend on the frequency as well.

*Electron Green functions.* The *bare* electron action is

$$S_e = \iint dx dx' \bar{\psi}_a(x)D_e^{ab}(x,x')\psi_b(x'), \qquad (B3)$$

where the causality structure of the inverse propagator is

$$\hat{D}_e = \begin{pmatrix} D_e^R & D_e^K \\ D_e^A & 0 \end{pmatrix}. \qquad (B4)$$

Here, $D_e^{R/A}(k) = \omega - \varepsilon_e(\mathbf{k}) \pm i0$ and $D_e^K(k) = 2i0F_e^{\text{th}}(\omega)$. The (infinitely small) Keldysh component in the non-interacting theory serves merely a role of a regularization, and is overshadowed by interaction as soon as they are included.

*Interaction.* The interaction $\hat{H}_{\text{int}}$ correspond to the action $S_{\text{int}} = -U \int_{\mathcal{C}} dx \bar{\phi}(x)\phi(x)\bar{\psi}(x)\psi(x)$, where $\mathcal{C}$ is the Keldysh contour that starts from $t = -\infty$, goes to $t = +\infty$ and returns to $t = -\infty$. To proceed we perform the Hubbard-Stratonovich transformation according to:

$$e^{iS_{\text{int}}} = \int \mathcal{D}\Delta \, e^{iS_\Delta + i\tilde{S}_{\text{int}}}, \qquad (B5)$$

where $S_\Delta = \int dx \bar{\Delta} U^{-1} \Delta$ and $\tilde{S}_{\text{int}} = \int dx [\bar{\phi}\bar{\psi}\Delta + \bar{\Delta}\psi\phi]$, where all the field are evaluated at $x$ and $\Delta$ is a fermionic field, since $\Delta \sim \phi\psi$. Now, we split the Keldysh contour into a forward (backward) branch going from $t = -\infty$ $(+\infty)$ to $t = +\infty$ $(-\infty)$, and we denote the fields by $\psi_+$, $\phi_+$ and $\Delta_+$ ($\psi_-$, $\phi_-$ and $\Delta_-$) residing on each branch. We next perform the Keldysh rotation [55], which brings the action to the following form

$$\tilde{S}_{\text{int}} = \int dx \, \frac{\gamma_{ab}^\alpha}{\sqrt{2}} \left[ \bar{\phi}^\alpha \bar{\psi}_a \Delta_b + \bar{\Delta}_a \psi_b \phi^\alpha \right]. \qquad (B6)$$

The formulas developed in this section, will be used to derive the self-energies in Sec. C. For completeness, the Keldysh causal structure for the GFs takes the form

$$\hat{G}_x = \begin{pmatrix} G_x^K & G_x^R \\ G_x^A & 0 \end{pmatrix} \qquad (B7)$$

for bosons and

$$\hat{G}_i = \begin{pmatrix} G_i^R & G_i^K \\ 0 & G_i^A \end{pmatrix} \qquad (B8)$$

for fermions ($i = e, \Delta$).

## Appendix C: Derivation of self-energies

After introducing the molecular field via the HS transformation, see Eq. (B5), the full action can be written as a sum of four components $S = S_x + S_e + S_\Delta + \tilde{S}_{\text{int}}$, as given in Sec. B. From this action we derive below the self-energies (SE) for the excitons, molecules and electrons.

### 1. Exciton SE

To begin, we integrate out the electrons from the path-integral generating functional $\mathcal{Z}$. That is, we write $e^{iS_{x\Delta}} = \int \mathcal{D}\psi e^{i(S_e + \tilde{S}_{\text{int}})}$, which results in the effective interaction between excitons and molecules.

$$S_{x\Delta} = -\int_{xx'} \frac{\gamma_{ab}^\alpha \gamma_{a'b'}^{\alpha'}}{2} \phi_\alpha(x)\bar{\Delta}_b(x)G_e^{aa'}(x,x')\bar{\phi}_{\alpha'}(x')\Delta_{b'}(x'), \qquad (C1)$$

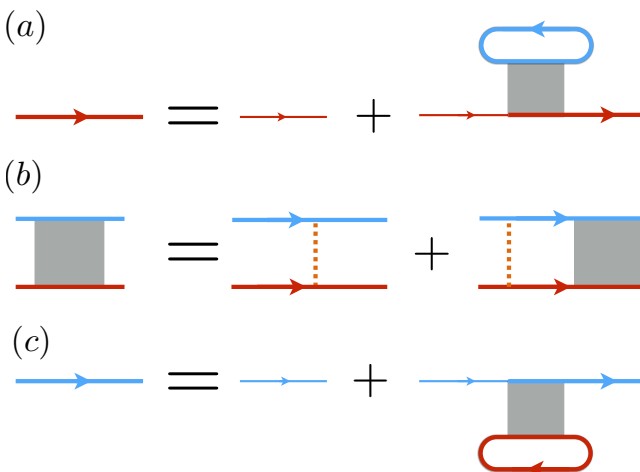

FIG. 5. The Dyson equations for (a) the exciton Green's function (red arrow), (b) molecular propagator (gray square is the $T$-matrix, i.e., $T = G_\Delta$) and (c) electron Green's function (blue arrow). The orange dotted line is the interaction, the bold arrows correspond to dressed propagators while the thin to the bare ones. We note that in our model with the contact potential, the $T$-matrix is a function of the total momentum of the two incoming lines. The Keldysh structure as well as the space-time or energy-momentum variables is not shown. For a detailed derivation, and how to transform the diagrams to mathematical expressions, see Ref. [37].

where $\int_x \equiv \int dx$ and $\int_{xx'} \equiv \int dx \int dx'$, etc.

To derive the exciton SE in the following step we integrate out the molecules. To this end, we rewrite

$$S_\Delta + S_{x\Delta} = \int_{xx'} \bar{\Delta}_b(x)B_{bb'}(x,x')\Delta_{b'}(x'), \qquad (C2)$$

where

$$B_{bb'}(x,x') = D_\Delta^{bb'}(x,x') - W^{bb'}(x,x'), \qquad (C3)$$

with $W^{bb'}(x,x') \equiv \frac{\gamma_{ab}^\alpha \gamma_{a'b'}^{\alpha'}}{2} \phi_\alpha(x)G_e^{aa'}(x,x')\bar{\phi}_{\alpha'}(x')$. Now we can integrate out molecules and introduce the action corresponding to the self-energy $S_x^{\text{SE}}$

$$e^{iS_x^{\text{SE}}} = \int \mathcal{D}\Delta e^{i(S_\Delta + S_{x\Delta})}. \qquad (C4)$$

Using the trace-log formula, i.e., $\det A = e^{\text{Tr}\ln A}$, we obtain

$$S_x^{\text{SE}} = -i\text{Tr}\ln\left(\hat{1} - \hat{G}_\Delta \circ \hat{W}\right) \approx i\text{Tr}\left(\hat{G}_\Delta \circ \hat{W}\right), \qquad (C5)$$

where $\circ$ here denotes matrix multiplication both in spacetime and Keldysh subspace, and in the last step we linearized in $W$. Employing now the explicit form of $\hat{W}$, we write $S_x + S_x^{\text{SE}}$ as a quadratic action $\int_{xx'} \bar{\phi}^\alpha(x)[D_x^{\alpha\alpha'}(xx') - \Sigma_x^{\alpha\alpha'}(xx')]\phi^{\alpha'}(x')$. In this way we arrive at the exciton SE shown in Eq. (A8b). We note, that in evaluations we should use bare GFs, but in the

self-consistent theory we can upgrade bare GFs to the dressed GFs in SEs. This can be derived from the perturbative diagrammatic expansion summing certain class of diagrams, or using the $\Phi$-functional [65]. The resulting Dyson equation is shown in Fig. 5a.

### 2. Molecule SE

To calculate the molecular self-energy, after tracing out the electron degrees of freedom, we integrate out the excitons. To this end, we write

$$S_x + S_{x\Delta} = \int_{xx'} \bar{\phi}(x) A_{\alpha\alpha'}(x, x') \phi_{\alpha'}(x'), \qquad (C6)$$

where $S_{x\Delta}$ is given in Eq. (C1), and

$$A_{\alpha\alpha'}(x, x') = D_x^{\alpha\alpha'}(x, x') - V^{\alpha\alpha'}(x, x'), \qquad (C7)$$

while the matrix $\hat{V}$ is expressed as

$$V^{\alpha\alpha'}(x, x') = \frac{\gamma_{ab}^{\alpha} \gamma_{a'b'}^{\alpha'}}{2} \bar{\Delta}_{b'}(x') G_e^{a'a}(x', x) \Delta_b(x). \quad (C8)$$

Now, we average over the exciton fields

$$e^{iS_{\Delta}^{\mathrm{SE}}} = \int \mathcal{D}\phi\, e^{i(S_x + S_{x\Delta})}, \qquad (C9)$$

and we define the effective action $S_{\Delta}^{\mathrm{SE}}$. Employing now the properties of Gaussian integrals, and the trace-log formula, we can write the action

$$S_{\Delta}^{\mathrm{SE}} = i\mathrm{Tr}\left[\hat{1} - \frac{1}{2}\hat{G}_x \circ \hat{V}\right] \approx -\frac{i}{2}\mathrm{Tr}\left[\hat{G}_x \circ \hat{V}\right], \quad (C10)$$

where in the final step we left only the linear term in $\hat{V}$. The action $S_{\Delta}^{\mathrm{SE}}$ together with $S_{\Delta}$ leads to the identification of the self-energy $\hat{\Sigma}_{\Delta}$ of the form, given in Eq. (A8a). Once again, similarly to the calculation of the exciton self-energy, the GFs in self-consistent calculations are upgraded to the dressed GFs in $\hat{\Sigma}_{\Delta}$. The diagrams contributing to the self-energy lead to the Dyson equation for the molecular GF shown in Fig. 5b.

### 3. Electron SE

The calculation of electron self-energy $\hat{\Sigma}_e$ proceeds similarly to the evaluation of $\hat{\Sigma}_x$, shown in Sec. C 1. The only difference is the use of Grassmann numbers instead of complex-valued fields and we average over exciton field after tracing out the molecular degrees of freedom. This approach leads to $\hat{\Sigma}_e$ as shown in Eq. A8c. The respective diagrammatic formulation for the Dyson equation is shown in Fig. 5c.

In principle, we should also add electron-electron interactions that would be responsible for the thermalisation

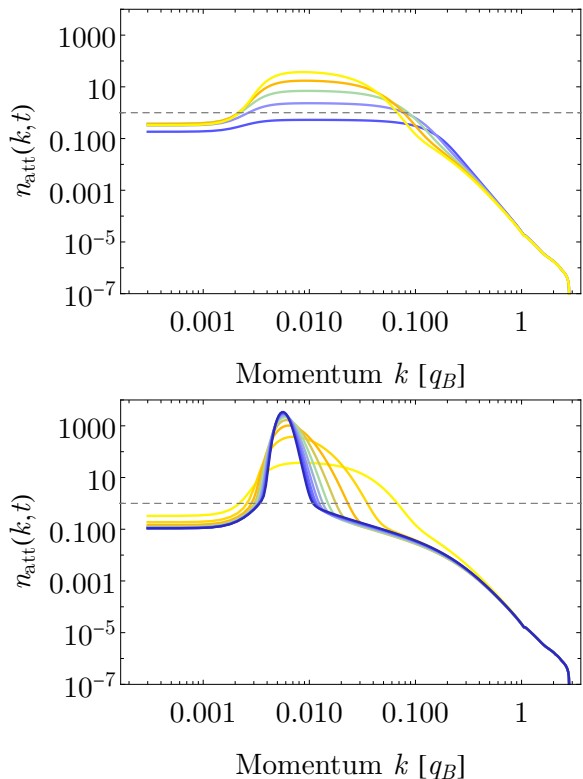

FIG. 6. The distribution function $n_{\mathrm{att}}(\mathbf{k}, t)$ for $t/\tau_{\mathrm{exc}} = 0.2$, 0.4, 0.6, 0.8, 1.0 (upper panel, curves from bottom to top) and $t/\tau_{\mathrm{exc}} = 1, 2, 3, \ldots, 9, 10$ (bottom panel, curves from bottom to top within the peak). The parameters correspond to the main panel of Fig. 1(c) of the main text. The dashed horizontal line indicates the threshold for stimulated scattering.

of electrons or a heat bath for electrons, that would dissipate the energy of the electrons excited during collisions with the excitons. In this work, however, we assume these thermalisation processes are very effective and rapidly cool the electron gas. Therefore, we assume that the electrons are kept in a thermal state.

### Appendix D: Time evolution of the distribution function.

In Fig. 6 we show the distribution function $n_{\mathrm{att}}(t)$ for $t/\tau_{\mathrm{exc}} = 0.2, 0.4, 0.6, 0.8, 1, 2, 3, \ldots, 9, 10$ of the time dynamics shown in the main panel of Fig. 1(c) in the main text. The evolution shows a fast stabilization of the high-energy population in the region close to the effective attractive polaron pump. Subsequently, a broad peak for $k_{\mathrm{rad}} \lesssim k \ll k_F$ grows, where $k_{\mathrm{rad}}$ is the effective width of the loss profile. When the population exceeds 1, the bosonic stimulation reshapes the distribution bringing most of the population to the tail of the light cone. At this point, the scattering from the higher-energy polarons to the peak is compensated by the radiative loss,

and the peak maintains its shape and position.

## Appendix E: Derivation of the relaxation rate $\Gamma_{\mathrm{att}}(\mathbf{k})$

Here we derive the formula for the relaxation rate $\Gamma_{\mathrm{att}}$. Our starting point is the formula for the transition rate. Replacing the sums with integrals, we obtain:

$$W_{\mathbf{k},\mathbf{k}'}^{\mathrm{att,att}} = \int \frac{d^2q}{(2\pi)^2} |T(\mathbf{q}+\mathbf{k}', \varepsilon_{\mathrm{att}}(\mathbf{k}') + \varepsilon_e(\mathbf{q}))|^2$$
$$\times 2\pi\delta(\varepsilon_{\mathrm{att}}(\mathbf{k}) - \varepsilon_{\mathrm{att}}(\mathbf{k}') + \varepsilon(\mathbf{q} - \delta\mathbf{k}) - \varepsilon_e(\mathbf{q}))$$
$$\times Z_{\mathrm{att}}(\mathbf{k}) Z_{\mathrm{att}}(\mathbf{k}') n_e(\mathbf{q})(1 - n_e(\mathbf{q} - \delta\mathbf{k})), \quad (E1)$$

where $\delta\mathbf{k} = \mathbf{k} - \mathbf{k}'$. In the following, since only the integral over $\mathbf{q}$ is important, for brevity, we write $Z \to Z_{\mathrm{att}}(\mathbf{k})$, $Z' \to Z_{\mathrm{att}}(\mathbf{k}')$, $\varepsilon \to \varepsilon_{\mathrm{att}}(\mathbf{k})$ and $\varepsilon' \to \varepsilon_{\mathrm{att}}(\mathbf{k}')$.

First, anticipating that the scattering takes place mainly around the Fermi surface, i.e., $|\mathbf{q}| \approx k_F$, for sufficiently small $|\mathbf{k}'| \ll k_F$, we may write

$$|T(\mathbf{q}+\mathbf{k}', \varepsilon'+\varepsilon_e(\mathbf{q}))|^2 \approx \left|T(\mathbf{q}, \varepsilon_{\mathrm{att}}(0)+E_F)|_{|\mathbf{q}|=k_F}\right|^2 \equiv |T|^2.$$
$$(E2)$$

Due to the cylindrical symmetry, the $T$ matrix depends only on $|\mathbf{q}| \approx k_F$. The rhs, which is now $\mathbf{q}$-independent we denote with $|T|^2$. We are thus left with the following integral

$$W_{\mathbf{k},\mathbf{k}'}^{\mathrm{att,att}} = ZZ'|T|^2 \int \frac{d^2q}{(2\pi)^2}$$
$$\times 2\pi\delta(\varepsilon - \varepsilon' + \varepsilon(\mathbf{q} - \delta\mathbf{k}) - \varepsilon_e(\mathbf{q}))$$
$$\times n_e(\mathbf{q})(1 - n_e(\mathbf{q} - \delta\mathbf{k})). \quad (E3)$$

Introducing the step function $n_e(\mathbf{q}) = \theta(E_F - \varepsilon_e(\mathbf{q}))$, and using the energy conservation the last line can be rewritten as

$$n_e(\mathbf{q})(1 - n_e(\mathbf{q} - \delta\mathbf{k})) = \quad (E4)$$
$$\theta(E_F - \varepsilon_e(\mathbf{q}))[1 - \theta(E_F - \varepsilon_e(\mathbf{q}) - \varepsilon + \varepsilon')]. \quad (E5)$$

At this point, it is convenient to introduce the following quantities:

$$\Delta\varepsilon \equiv \varepsilon - \varepsilon' + \delta\mathbf{k}^2/2m_e \quad (E6)$$
$$\Delta v \equiv |\delta\mathbf{k}|/m_e. \quad (E7)$$

Finally, we obtain the following form of the transition rate:

$$W_{\mathbf{k},\mathbf{k}'}^{\mathrm{att,att}} = ZZ'|T|^2 \int_0^\infty qdq \int_{-\pi}^\pi \frac{d\phi}{(2\pi)}$$
$$\times \frac{1}{|\Delta v \cos\phi|} \delta\left(\frac{\Delta\varepsilon}{\Delta v \cos\phi} - q\right)$$
$$\times \theta(E_F - \varepsilon_e(\mathbf{q}))[1 - \theta(E_F - \varepsilon_e(\mathbf{q}) - \varepsilon + \varepsilon')], \quad (E8)$$

where $\phi$ is the angle between $\delta\mathbf{k}$ and $\mathbf{q}$. The energetic delta ensures the length of $|\mathbf{q}| = q(\phi) = \frac{\Delta\varepsilon}{\Delta v \cos\phi} \geqslant 0$. If $\Delta\varepsilon/\Delta v$ is small, only the angles around $\phi \approx \pm\pi/2$ are contributing to the integral over $\phi$.

Anticipating that $q \approx k_F$, we may approximate

$$W_{\mathbf{k},\mathbf{k}'}^{\mathrm{att,att}} = ZZ'|T|^2 \frac{k_F^2}{\Delta\varepsilon} \int \frac{d\phi}{2\pi} f_e(1 - f_e'), \quad (E9)$$

where $f_e = \theta(E_F - \varepsilon_e(\mathbf{q}))|_{q=q(\phi)}$ and $f_e' = \theta(E_F - \varepsilon_e(\mathbf{q}) - \varepsilon + \varepsilon')|_{q=q(\phi)}$. From this formula, we see that it is non-vanishing only if $\varepsilon < \varepsilon'$, i.e., only in the case of cooling.

Now, we focus only on $\phi \approx \pi/2$, and we multiply the result by 2. The integral in Eq. (E9) is a length of the curve given by $q(\phi)$ with the constraint that $\varepsilon_e(\mathbf{q})$ should lie in a thin shell $E_F - (\varepsilon' - \varepsilon) < \varepsilon_e(\mathbf{q}) < E_F$. In changing the angle $\phi$ by $\Delta\phi$ the electron changes its energy by $\Delta\varepsilon_e$:

$$\Delta\varepsilon_e(q) = \frac{1}{2m_e}\Delta\left(\frac{\Delta\varepsilon}{\Delta v \cos\phi}\right)^2 = \frac{\Delta\varepsilon^2 \sin\phi\Delta\phi}{m_e\Delta v^2 \cos^3\phi}. \quad (E10)$$

Since $\sin\phi \approx 1$ and $q(\phi) \approx k_F$ we obtain:

$$\Delta\varepsilon_e(q) \approx \frac{\Delta v}{\Delta\varepsilon}\frac{k_F^3}{m_e}\Delta\phi. \quad (E11)$$

Since the maximum change of the electrons' energy that is compatible with the energy conservation is given by $\Delta\varepsilon_e(q) = \varepsilon' - \varepsilon$, we obtain

$$\Delta\phi \approx (\varepsilon' - \varepsilon)\frac{\Delta\varepsilon}{\Delta v}\frac{m_e}{k_F^3}. \quad (E12)$$

Therefore, the integral in Eq. (E9) is

$$\int d\phi f_e(1 - f_e') \approx 2\Delta\phi, \quad (E13)$$

where the factor of 2 takes into account the contribution from $\phi \approx -\pi/2$.

Finally, we obtain the transition rates:

$$W_{\mathbf{k},\mathbf{k}'}^{\mathrm{att,att}} = Z(\mathbf{k})Z(\mathbf{k}')|T(k_F, E_F + \varepsilon_{\mathrm{att}}(0))|^2$$
$$\times \frac{1}{\pi}\frac{m_e}{k_F\Delta v}(\varepsilon_{\mathrm{att}}(\mathbf{k}') - \varepsilon_{\mathrm{att}}(\mathbf{k}))$$
$$\times \theta(\varepsilon_{\mathrm{att}}(\mathbf{k}') - \varepsilon_{\mathrm{att}}(\mathbf{k})). \quad (E14)$$

In the next step, we derive the integral over the angle between $\mathbf{k}$ and $\mathbf{k}'$, which we denote with $\phi'$. Since it enters only in $1/\Delta v$, we may write

$$\int \frac{d\phi'}{2\pi}\frac{1}{\Delta v} = \frac{2m_e}{\pi}\frac{K\left(\frac{4kk'}{(k+k')^2}\right)}{k+k'}, \quad (E15)$$

where $k = |\mathbf{k}|$, $k' = |\mathbf{k}'|$ and $K(x)$ is the complete elliptic integral of the first kind. Using the identity

$$\int_0^1 dx x(1-x)K\left(\frac{4x}{(1+x)^2}\right) = \frac{4}{9} \quad (E16)$$

we may directly evaluate the relaxation rate

$$\Gamma_{\mathrm{att}}(\mathbf{k}') = \int_0^{k'} \frac{kdk}{2\pi} \int \frac{d\phi'}{2\pi} W_{\mathbf{k},\mathbf{k}'}^{\mathrm{att,att}}$$

$$\approx \frac{m_e^2 Z_{\mathrm{att}}^2(0)|T(k_F, \varepsilon_{\mathrm{att}}(0) + E_F)|^2}{m_{\mathrm{att}} k_F} \frac{2}{9\pi^3}|\mathbf{k'}|^3, \text{ (E17)}$$

where we have used Eq. (E14), approximated $Z_{\mathrm{att}}(\mathbf{k}) \approx Z_{\mathrm{att}}(0)$ (also for $\mathbf{k'}$), and assumed effective mass of the polarons: $\varepsilon_{\mathrm{att}}(\mathbf{k'}) - \varepsilon_{\mathrm{att}}(\mathbf{k}) = (\mathbf{k'}^2 - \mathbf{k}^2)/2m_{\mathrm{att}}$. We notice that $\Gamma_{\mathrm{att}}(\mathbf{k}) \propto |\mathbf{k}|^3$ for $|\mathbf{k}| \ll k_F$.

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
