# Peer review of "Fermi polaron laser in two-dimensional semiconductors"

_SciPost Physics_

## Round 2 · Referee Report · Anonymous (Referee 1) · 2022-5-29

Report

The manuscript by Wasak et al. presents calculations of the dynamics of populations in a system of exciton polarons. These calculations, following from deriving a quantum Boltzmann equation for the polaron populations, describe relaxation from a population of polarons in the (upper) repulsive state to the (lower) attractive state, and then relaxation within the attractive branch. The main result of the manuscript is that this system shows nonlinear behavior of the relaxation process, due to stimulated scattering, analogous to a laser. Key to this is that relaxation from the repulsive to the attractive branch involves transitions to higher momentum states, due to energy-momentum conservation in scattering with electrons. This means that these attractive branch polarons start outside the light cone. When the population is low, most exciton-polarons are lost by non-radiative decay before they relax into the light cone. When population is high, stimulated scattering means that most exciton-polarons relax to the light cone. This leads to a laser-like transition in the radiative efficiency.

The work appears novel and in general the methods seem appropriate (see specific questions below). On balance, there is a reasonable case that this work "opens a new pathway in an existing or a new research direction, with clear potential for multipronged follow-up work", consistent with the SciPost Physics criteria. This is because it shows the potential for interesting results in studying the dynamics of this polaron system, and its response to selectively pumping the repulsive branch. While there has been significant recent interest on this exciton-polaron system, this has not generally explored the dynamics or nonlinear response to pumping.

This current work does rely on some approximations that may require further investigation. For example, the role of the trion-hole continuum is mostly ignored, and effects of broadening due to coupling between polaron resonances and the continuum is not considered in the dynamics. Further, relaxation is assumed to only occur via the excitonic part of the polaron scattering from electrons, neglecting the role of electron-electron scattering on the hybrid polaron. Nevertheless, despite these issues (which future work might address), this manuscript does provide reasonable evidence that interesting behavior should arise in this system, prompting future work.

As noted below, there are a number of issues that require clarifying in the current manuscript.

Requested changes

I list first here both points that require addressing before publication, and then some optional suggestions the authors may wish to consider.

  1. I have some concern about describing the behavior as being a laser, rather than laser-like. I agree there is a strong similarity, in that the transition is driven by a change of radiative efficiency. However the behavior is not driven by light amplification nor by stimulated emission of radiation. The term laser-like may be clearer; the authors may wish to consider whether they think the current term is appropriate.

  2. I note that the references in the manuscript generally appear to date from before the initial submission of this paper in March 2021. The authors may wish to consider whether there are more recent references that should be included. For example, work on other aspects of interactions between polarons and electrons such as 10.1103/PhysRevB.103.075417 and 10.1103/PhysRevB.105.L041401

  3. Regarding Eq. 9, the authors may wish to consider including a figure that shows the comparison between this analytic prediction and the results of the numerics for the form of the tail.

  4. In section VI, it is stated that "the linewidth rapidly increases as the radiative loss becomes more prominent" at large pump strength. While this seems plausible from the physics described, it is hard to see how this appears in the expression given for S(omega). Could the authors explain where this appears in the equation more clearly?

  5. In defining g^{(1)}(r,t), the definition written appears to suggest this is just the phase of G^{(1)}(r,t). I would assume the definition should be G^{(1)}(r,t)/|G^{(1)}(0,0)|

6a. In the start of Appendix A (after Eq. A2) A few points are unclear about the model of pumping. First, it is not clear why the pump term consists of both a term that increases polaron population and one that decreases it. Could this be explained further?

6b. Second, it is stated here that the assumption of frequency-independent pump is relaxed later. While it is clear in the main text that this is so (in that only the repulsive branch is pumped), the relaxation of this assumption does not seem to be explicitly discussed in the appendix. Should this be discussed in Appendix B, when writing the pumping term in the bare Green's functions?

  1. Before Eq. A6, "Habbard"->"Hubbard"

  2. The diagrams shown in Figure 5 do not seem to correspond directly to the equations actually used, as given in Eq. A8. Most notably,in the equations, the molecular channel is treated as a specific resonance, leading to a Green's function with a single argument. In contrast, the diagrams suggest it is dealt with by a T matrix, which would be a function of three momenta, that includes both the bound molecular channels and the scattering continuum. The equations written would seem instead to correspond to those in the attached file, where red and blue lines are as in the manuscript and grey lines indicate the molecule Green's function.

This should be clarified before publication.

  1. In Appendix A.5, the sentence "In Fig. 1(b) in the main text..." is unclear. I suggest this should read "In Fig.1(b) in the main text we present the results of the self-consistent calculations of the excitation spectrum, using the parameters as given above".

  2. In Appendix A.6 "These function are" -> "These functions are"

  3. In Appendix A.6, in discussing Z_alpha(k), the method described of dividing the energy range by the maximum of the molecular spectral function presumably has the effect of imposing that Z_att+Z_rep=1. That is, this assumes there is no transfer of spectral weight to the trion-hole continuum. Is this assumption justified?

  4. Throughout Appendix A7 and Appendices B, C, the manuscript keeps switching whether to label real-space coordinates as r or x. Given that x is used for the exciton channel label, it might be clearer to use r throughout. In any case, the notation should be consistent.

  5. After Eq. A12, the discussion of momentum arguments is confusing, and seems likely to be wrong. The equation involves three momenta, k, q, q', but the discussion refers also to a fourth momentum p. This should be checked and clarified.

I also note that in A12, each line has a separate equation number, even though this is only one equation. The same applies to A15. (Other equations, such as E1, E8 do not do this.

  1. In Figure 6, it would be helpful to add a legend or colorscale to label the meaning of the line colors.

  2. In Appendix E, the subscript e on \epsilon_e is missing in Eq. E1, Eq. E3.

  3. I Appendix E, after E14, the discussion changes from calculating W to calculating Gamma. However the text is not very clear. It would help to change "In the next step" to instead say "Having now calculated $W...$ we now turn to calculating $\Gamma....$" or equivalent.

---

## Round 2 · Referee Report · Anonymous (Referee 1) · 2022-5-29

Report

The manuscript by Wasak et al. presents calculations of the dynamics of populations in a system of exciton polarons. These calculations, following from deriving a quantum Boltzmann equation for the polaron populations, describe relaxation from a population of polarons in the (upper) repulsive state to the (lower) attractive state, and then relaxation within the attractive branch. The main result of the manuscript is that this system shows nonlinear behavior of the relaxation process, due to stimulated scattering, analogous to a laser. Key to this is that relaxation from the repulsive to the attractive branch involves transitions to higher momentum states, due to energy-momentum conservation in scattering with electrons. This means that these attractive branch polarons start outside the light cone. When the population is low, most exciton-polarons are lost by non-radiative decay before they relax into the light cone. When population is high, stimulated scattering means that most exciton-polarons relax to the light cone. This leads to a laser-like transition in the radiative efficiency.

The work appears novel and in general the methods seem appropriate (see specific questions below). On balance, there is a reasonable case that this work "opens a new pathway in an existing or a new research direction, with clear potential for multipronged follow-up work", consistent with the SciPost Physics criteria. This is because it shows the potential for interesting results in studying the dynamics of this polaron system, and its response to selectively pumping the repulsive branch. While there has been significant recent interest on this exciton-polaron system, this has not generally explored the dynamics or nonlinear response to pumping.

This current work does rely on some approximations that may require further investigation. For example, the role of the trion-hole continuum is mostly ignored, and effects of broadening due to coupling between polaron resonances and the continuum is not considered in the dynamics. Further, relaxation is assumed to only occur via the excitonic part of the polaron scattering from electrons, neglecting the role of electron-electron scattering on the hybrid polaron. Nevertheless, despite these issues (which future work might address), this manuscript does provide reasonable evidence that interesting behavior should arise in this system, prompting future work.

As noted below, there are a number of issues that require clarifying in the current manuscript.

Requested changes

I list first here both points that require addressing before publication, and then some optional suggestions the authors may wish to consider.

  1. I have some concern about describing the behavior as being a laser, rather than laser-like. I agree there is a strong similarity, in that the transition is driven by a change of radiative efficiency. However the behavior is not driven by light amplification nor by stimulated emission of radiation. The term laser-like may be clearer; the authors may wish to consider whether they think the current term is appropriate.

  2. I note that the references in the manuscript generally appear to date from before the initial submission of this paper in March 2021. The authors may wish to consider whether there are more recent references that should be included. For example, work on other aspects of interactions between polarons and electrons such as 10.1103/PhysRevB.103.075417 and 10.1103/PhysRevB.105.L041401

  3. Regarding Eq. 9, the authors may wish to consider including a figure that shows the comparison between this analytic prediction and the results of the numerics for the form of the tail.

  4. In section VI, it is stated that "the linewidth rapidly increases as the radiative loss becomes more prominent" at large pump strength. While this seems plausible from the physics described, it is hard to see how this appears in the expression given for S(omega). Could the authors explain where this appears in the equation more clearly?

  5. In defining g^{(1)}(r,t), the definition written appears to suggest this is just the phase of G^{(1)}(r,t). I would assume the definition should be G^{(1)}(r,t)/|G^{(1)}(0,0)|

6a. In the start of Appendix A (after Eq. A2) A few points are unclear about the model of pumping. First, it is not clear why the pump term consists of both a term that increases polaron population and one that decreases it. Could this be explained further?

6b. Second, it is stated here that the assumption of frequency-independent pump is relaxed later. While it is clear in the main text that this is so (in that only the repulsive branch is pumped), the relaxation of this assumption does not seem to be explicitly discussed in the appendix. Should this be discussed in Appendix B, when writing the pumping term in the bare Green's functions?

  1. Before Eq. A6, "Habbard"->"Hubbard"

  2. The diagrams shown in Figure 5 do not seem to correspond directly to the equations actually used, as given in Eq. A8. Most notably,in the equations, the molecular channel is treated as a specific resonance, leading to a Green's function with a single argument. In contrast, the diagrams suggest it is dealt with by a T matrix, which would be a function of three momenta, that includes both the bound molecular channels and the scattering continuum. The equations written would seem instead to correspond to those in the attached file, where red and blue lines are as in the manuscript and grey lines indicate the molecule Green's function.

This should be clarified before publication.

  1. In Appendix A.5, the sentence "In Fig. 1(b) in the main text..." is unclear. I suggest this should read "In Fig.1(b) in the main text we present the results of the self-consistent calculations of the excitation spectrum, using the parameters as given above".

  2. In Appendix A.6 "These function are" -> "These functions are"

  3. In Appendix A.6, in discussing Z_alpha(k), the method described of dividing the energy range by the maximum of the molecular spectral function presumably has the effect of imposing that Z_att+Z_rep=1. That is, this assumes there is no transfer of spectral weight to the trion-hole continuum. Is this assumption justified?

  4. Throughout Appendix A7 and Appendices B, C, the manuscript keeps switching whether to label real-space coordinates as r or x. Given that x is used for the exciton channel label, it might be clearer to use r throughout. In any case, the notation should be consistent.

  5. After Eq. A12, the discussion of momentum arguments is confusing, and seems likely to be wrong. The equation involves three momenta, k, q, q', but the discussion refers also to a fourth momentum p. This should be checked and clarified.

I also note that in A12, each line has a separate equation number, even though this is only one equation. The same applies to A15. (Other equations, such as E1, E8 do not do this.

  1. In Figure 6, it would be helpful to add a legend or colorscale to label the meaning of the line colors.

  2. In Appendix E, the subscript e on \epsilon_e is missing in Eq. E1, Eq. E3.

  3. I Appendix E, after E14, the discussion changes from calculating W to calculating Gamma. However the text is not very clear. It would help to change "In the next step" to instead say "Having now calculated $W...$ we now turn to calculating $\Gamma....$" or equivalent.

Attachment

---

## Round 2 · Referee Report · Anonymous (Referee 2) · 2022-7-5

Report

The paper “Fermi polaron laser in two-dimensional semiconductors” studies the relaxation dynamics of a charged TMD monolayer in the Fermi polaron regime driven by an external laser which is resonantly tuned to the repulsive branch at zero momentum. The main result of the paper is the finding of a lasing transition when increasing the pump power in the regime of weak coupling to light. The transition is due to the decay from the repulsive to the attractive branch at finite momentum, and a subsequent slow relaxation to a state at the edge of the light cone because of exciton-electron collisions. The lasing transition is associated to an extended spatial and temporal first order coherence, with a coherent peak having a linewidth below the limit set by the exciton non-radiative lifetime.

The paper is certainly interesting, and it might deserve publication in some form, if the authors can address the comments listed below. The finding of a lasing transition in the Fermi polaron regime for charged monolayers is an interesting outcome. However, the importance of the implications of this findings are not clear and are feeble. The presence of a Fermi sea is necessary for allowing relaxation in the attractive branch because of electron-exciton scattering. However, the efficiency of this scattering as a function of the electron density has not been established. The authors consider a fixed value of the Fermi sea energy EF, while they could easily conduct a study as a function of this parameter. Further, this lasing transition seems to have a relatively low threshold. However, the dependence of the threshold from EF has not been assessed. At the same time, the paper is overall very laborious to read. The main paper contains almost only the results derived numerically. However, the formalism, as well as the derivation of the kinetic equations and time evolution of the distribution function, are entirely contained in the appendices together with the many approximations that have been carried on. This makes the paper difficult to read and it makes it difficult to establish the soundness and physical origin of the many approximations that have been carried on. Further, I found much overlap between this work and Ref. [37] “Quantum-Zeno Fermi polaron in the strong dissipation limit” from some of the authors --- which is presumably the reason why all the formalism is relegated to the appendix as not entirely original? In this case, the authors should state this clearly, and discuss in detail the differences between the current and previous work.

Requested changes

Specific comments:

1. Introduction & conclusions. The introduction refers to the semiconductor realisation of the polaron regime, specifically to the case of TMD monolayers. However, in the first paragraph of the introduction, when referring to attractive and repulsive polaron branches, the authors cite Refs. [21-32], which also include few references from the cold atom literature on the Fermi polaron problem. I would separate the discussion of the two cases, as this would very much help the reader to comprehend the analogies and differences between the two realisations of the polaron problem. It is not clear how this work results would also apply to the cold atom case --- i.e., in that case, can one resonantly excite the repulsive branch? In the same spirit, the sentence in the conclusions “the richness of the underlying nonequilibrium dynamics sheds light on the quantum many-body nature of Fermi polarons, both in solid-state materials [33] and ultracold atomic gases [5-54]” is very vague. The author should argue why their calculations are relevant for the cold atom realisation.
2. Study for different values of the electron gas density. The authors work at a fixed value of the electron gas density, specifically fixing the Fermi sea energy EF to the trion binding energy EB. It would be interesting (and not very costly) to study the evolution of the lasing transition with EF. Presumably the exciton-electron collisions become less efficient for lower values of the electron density, leading to an increase of the lasing threshold? How does the spatiotemporal coherence change? Does the oscillator strength transfer from the repulsive to the attractive branch with EF play any role in the predicted lasing transition? I believe this work would strongly benefit from such a study, without requiring much effort from the author’s side.
3. At the same time, the experimental regime where the Fermi energy equals the trion binding energy is not particularly well described by the model considered by the authors, where the exciton is assumed to be tightly bound and structureless. In fact, for these values of EF, this model is shown to underestimate collision effects and the increase of the linewidth of the repulsive branch with EF is very much underestimated compared to experiments. This point should be addressed by the authors, which should justify the validity of the model. One additional reason to study the dependence of the results from EF and push this study to lower values of EF, where the use of the model is justified.
4. Independent polaron assumption. One of the main assumptions of this work is that pumping generates a low density of polaron so that they can be treated independently, and one can ignore the collisions between polarons. It is not clear to me this assumption is respected at the lasing transition at high pump powers.
5. Temperature effects. The authors are dismissive of temperature effects. In particular, in Sec. III, they state that “In TMDs at cryogenic temperatures we have EB=10^2T and we henceforth set T = 0, thereby ignoring transitions from the attractive to the repulsive branch.” It is not clear what is meant for transitions from the attractive to the repulsive branch. Aside this, temperature can have important effects on the linewidth of the attractive branch and thus can considerably affect the laser transition predicted in this work. These aspects should be analysed and discussed in detail.
6. Experimental accessibility of the momentum spectrum. The intensity peak at non-zero momentum close to the light cone edge in the late time momentum distribution function is one of the main results of this work which provides evidence for stimulated scattering --- see Fig. 2(a). Authors should discuss whether this part of the spectrum at finite momentum is experimentally accessible in the weak coupling regime.
7. Lasing transition threshold. At the end of section V, the authors discuss that the lasing threshold for the parameters considered is quite low. As commented above, the dependence of the threshold from EF is an important aspect of this study. At the same time, they compare their threshold with the case of a TMD monolayer nanocavity of Ref. [40] which dates back 2015. Even lower thresholds have been found recently in the strong coupling polaritonic regime, see e.g., “Ultralow Threshold Polariton Condensate in a Monolayer Semiconductor Microcavity at Room Temperature” [Nano Lett. 2021, 21, 3331−3339], where a power density off 0.06W/cm^2 has been reported. The authors should put their result into an updated context. Further, how they expect their results to change in the strong coupling regime?
8. Polaron-molecule transition. In the cold atom context, a polaron-molecule transition has been predicted at low EF --- though has not been observed for the semiconductor case. Can scattering to the molecular branch affect the author results?
9. Appendix A. There is no need to repeat the form of the Hamiltonian in equation (A1) describing the exciton-electron model used in the paper, as this is clear enough from the main text around equation (1). Appendices are not Supplemental Material. Instead, the authors should explain why all formalism, as well as the derivation of the kinetic equations and time evolution of the distribution function, are relegated to the appendices and do not belong to the main text. The impression is that this is because these results are not new, and the formalism coincides with those developed in Ref. [37] by some of the authors. In subsection 1 of this appendix, the authors state that “the formalism that we employ in this work closely follows the one developed in Ref. [37]”. What is new respect to that work should be explained in detail. Also, it is not clear why the authors describe first the master equation as a way to describe loss and the external drive of a laser, when instead they later resort to a Keldysh description. I would delete the entire introductory part of Appendix A and start directly with the subsection A1.
10. Appendix A5 Spectral function. In absence of non-equilibrium effects of pump and decay, I understand that the formalism developed in this work is equivalent to the Chevy Ansatz of a single particle-hole dressing of the Fermi sea, or equivalently to a T-matrix approach. This is the limit in which is evaluated the Fig. 1(b). If this is correct, it should be clearly explained in the Appendix as well as the main text.
11. Appendix A7 and validity of the Fermi liquid theory expressions. I would write a numbered equation for the expression of the renormalised exciton energy, so that one can refer to it when, later, the authors explain how one can derive the energies of attractive and repulsive polaron branches as zeros of the mass function. Further, it should be noted that the expressions for spectral weights and linewidths of Fermi liquid theory cannot be applied to this Fermi polaron problem. While it is true that the peaks of the spectral function corresponding to attractive and repulsive branches coincide with the zeros of the mass function, the repulsive branch is better described as a “continuum resonance”. In particular, its linewidth increases with EF and cannot be described as the imaginary part of the exciton self-energy. Similarly, the spectral weights cannot be evaluated from the inverse derivative of the renormalised exciton energy, rather, one must evaluate the area under the corresponding peaks in the spectral function. This should be clearly discussed.
12. Appendix E. Are the analytical estimates in this appendix relevant for the numerical results in the main text? If so, it should be analysed and explained.
13. Minor comments and typos
a. Appendix A1. “Habbard-Stratonovich transform” should be “Hubbard-Stratonovich transformation”.
b. Fig. 6. Add a legend as it is not immediately clear which times correspond to which colours in the two panels (as time increases from dark to light in top panel and opposite in bottom).
c. Fig. 2(a): add arrow or a legend to specify that Omega increases from red to blue colours.
d. Fig. 3: the y-log scale in the inset implies they are plotting |g^(1)|

---

## Editorial Decision

awaiting_resubmission